# CLAVATA3 mediated simultaneous control of transcriptional and post-translational processes provides robustness to the WUSCHEL gradient

Alexander Plong[1], Kevin Rodriguez[1], Mark Alber[2,3], Weitao Chen [2,3 ✉] & G. Venugopala Reddy [1,3 ✉]

Regulation of the homeodomain transcription factor WUSCHEL concentration is critical for stem cell homeostasis in *Arabidopsis* shoot apical meristems. WUSCHEL regulates the transcription of *CLAVATA3* through a concentration-dependent activation-repression switch. *CLAVATA3*, a secreted peptide, activates receptor kinase signaling to repress *WUSCHEL* transcription. Considering the revised regulation, *CLAVATA3* mediated repression of *WUSCHEL* transcription alone will lead to an unstable system. Here we show that *CLAVATA3* signaling regulates nuclear-cytoplasmic partitioning of *WUSCHEL* to control nuclear levels and its diffusion into adjacent cells. Our work also reveals that WUSCHEL directly interacts with EXPORTINS via EAR-like domain which is also required for destabilizing WUSCHEL in the cytoplasm. We develop a combined experimental and computational modeling approach that integrates *CLAVATA3*-mediated transcriptional repression of *WUSCHEL* and post-translational control of nuclear levels with the WUSCHEL concentration-dependent regulation of *CLAVATA3*. We show that the dual control by the same signal forms a seamless connection between de novo WUSCHEL synthesis and sub-cellular partitioning in providing robustness to the WUSCHEL gradient.

[1] Department of Botany and Plant Sciences, University of California Riverside, Riverside, CA 92521, USA. [2] Department of Mathematics, University of California Riverside, Riverside, CA 92521, USA. [3] Interdisciplinary Center for Quantitative Modeling in Biology, University of California Riverside, Riverside, CA 92521, USA. ✉email: weitaoc@ucr.edu; venug@ucr.edu

The precise spatiotemporal regulation of gene expression, growth, and differentiation are critical for stem cell maintenance. In the *Arabidopsis* shoot apical meristem (SAM), WUSCHEL (WUS), a homeodomain-containing transcription factor (TF), accumulates at a higher level at the site of synthesis, the rib meristem (RM), then diffuses into neighboring cells forming a concentration gradient[1–4]. WUS represses the expression of differentiation-promoting factors in the central zone (CZ), thus maintaining the stem cells[5,6]. As the stem cell descendants are displaced away from the influence of the WUS gradient, they begin to differentiate into leaves or flowers in the peripheral zone (PZ), and cells located beneath the RM, referred to as the basal L3 layer, differentiate into the stem[7]. The precise regulation of the WUS gradient is critical for stem cell maintenance and timely differentiation of stem cell progeny. The regulation of nuclear levels of WUS involves a balance between nuclear retention and nuclear export[8].

WUS activates and represses *CLAVATA3* (*CLV3*) transcription at lower and higher levels, respectively, by binding to the same *cis*-regulatory module (CRM) of the *CLV3* promoter[9]. CLV3, a secreted and processed peptide, signals through a receptor kinase signaling pathway to repress and restrict *WUS* expression to the RM[10–13]. Given this scenario, the earlier proposed model of CLV3 repressing *WUS* transcription will lead to an unstable system as elevated levels of WUS will repress *CLV3* and the resulting reduced CLV3 levels will, in turn, continue to elevate *WUS* expression leading to a higher number of stem cells. Similarly, a system with depleted levels of WUS activating *CLV3* will further downregulate *WUS* expression leading to further depletion of WUS and stem cell population. In those extreme cases, the classical WUS-CLV3 feedback model fails to maintain a robust accumulation of WUS under the new understanding of the WUS concentration-dependent regulation of *CLV3*. Loss of *WUS* repression in *clv3-2* null mutants leads to increased WUS protein accumulation in the RM/L3 layer due to an increase in expression and expansion of the *WUS* expression domain into L2 layer. However, WUS protein in the L1 layer fails to accumulate at a higher level despite a higher synthesis of the protein in the underlying cells[9,12,13], suggesting that CLV3-signaling could play an additional role in the maintenance of the WUS protein in the CZ.

Here we show that WUS physically interacts with EXPORTIN proteins by using the EAR-like motif which functions as a nuclear export signal (NES) and is also required for WUS destabilization in the cytoplasm. CLV3-signaling functions to offset the nuclear export of WUS in outer cell layers in addition to its role in repressing the *WUS* transcription in the RM. To gain a quantitative understanding of the dynamics of the coupled dual function of CLV3 in regulating the WUS gradient, we developed computational models that integrate layer-specific signals with WUS synthesis, subcellular distribution, diffusion, and degradation across cell layers with concentration-dependent reversible regulation of *CLV3* transcription. Our model predictions and the experimental analysis reveal that the simultaneous control of WUS synthesis and subcellular partitioning by the CLV3 regulates nuclear WUS concentration which, in turn, regulates *CLV3* levels, thus forming a dynamic feedback mechanism in the robust maintenance of the WUS protein gradient. Simultaneous control of the synthesis and nuclear concentration through the nuclear-cytoplasmic partitioning provides a seamless connection between the de novo WUS synthesis and replenishment of stem cells with new WUS protein molecules to sustain *CLV3* expression and stem cell homeostasis.

effect of CLV3-signaling on the WUS protein accumulation, we monitored the WUS protein reporter (*pWUS::eGFP-WUS*) in wild type and *clv3-2* mutant backgrounds in response to exogenous application of the bioactive CLV3 peptide (MCLV3) that has been previously shown to suppress *WUS* expression and SAM size[14]. If CLV3-signaling only represses *WUS* transcription, the MCLV3 application should lead to a rapid reduction of the WUS protein levels since instantaneous inhibition of *WUS* transcription was observed within 30 min of treatment[15]. Wild-type seedlings treated with 1 µM of the MCLV3, initially revealed a reduction in WUS protein accumulation within 3 h of MCLV3 treatment, while the protein levels were restored to levels observed at the initial time point and this level was maintained even after 6 (Fig. 1a–c, g), 24, 48, and 72 hr of MCLV3 treatment (Fig. 2a–d, k). In contrast, the *clv3-2* mutant SAMs treated with MCLV3 revealed a more drastic reduction in WUS levels in the L3 layer, but an increase in the L1 layer and this trend was consistently observed over 3, 6 (Fig. 1d–f, h), 24, and 48 hr of MCLV3 treatment (Fig. 2e–h, l). The inactive scrambled peptide (sCLV3) treatments did not affect *WUS* transcript (Supplementary Fig. 1e) and protein accumulation (Supplementary Fig. 1a, b). The observed reduction of WUS protein levels in the L3 layers of wild type and *clv3-2* SAMs upon MCLV3 treatment is likely due to reduced WUS protein synthesis caused by the repression of *WUS* transcription (Fig. 1i–k). Moreover, even after prolonged MCLV3 treatments extending up to 72 hr, WUS protein continued to accumulate in SAMs despite a severe downregulation of *WUS* transcript levels (Fig. 2i, j). These results suggested a new role for CLV3-signaling in regulating the nuclear levels and stability of the WUS protein.

To unambiguously test the role of CLV3-signaling in regulating the WUS protein levels, we uncoupled the effects of MCLV3 on the WUS protein from that of the *WUS* promoter by using dexamethasone (Dex)-inducible fluorescently tagged WUS reporter (*p35S::eGFP-WUS-GR*) expressed under the control of a heterologous ubiquitous promoter[8]. An earlier study has shown that the nuclear translocation of the WUS protein upon Dex application in wild-type plants leads to a staggered loss of the WUS protein in the central part of the SAM starting in the CZ between 6–12 h and reaching the lateral edges of the PZ by 24 h of Dex application showing WUS protein degradation after Dex-induced nuclear translocation (Fig. 3a–c)[8]. In *clv3-2* mutants, rapid degradation of the WUS protein could be seen within 3–6 h (Fig. 3f–h) of Dex application and a more widespread degradation was observed within 24 h of Dex application (Supplementary Fig. 2). However, the co-treatment with MCLV3 and Dex resulted in a stable nuclear accumulation of the WUS protein (Fig. 3d, i, k, l). The inactive scrambled peptide (sCLV3) and Dex treatments did not stabilize the WUS protein (Supplementary Fig. 1c, d). In rare cases, nuclear accumulation of eGFP-WUS-GR was observed upon MCLV3 treatment in DEX-untreated plants (Supplementary Fig. 3a′, b′) which could be due to the leakiness of the GR fusion. The *p35S::eGFP-GR* controls accumulated stably in the nuclei of wild type (Supplementary Fig. 4a–c, i), and *clv3-2* (Supplementary Fig. 4e–g, j) plants upon Dex treatments and MCLV3 treatments (Supplementary Fig. 4d, h) did not change its accumulation pattern showing that the instability associated with the *p35S::eGFP-WUS-GR* upon dex-induced nuclear translocation and its subsequent stabilization by MCLV3 is specific to the WUS protein. Taken together, these results suggested that CLV3-signaling either directly promotes WUS stability or blocks nuclear export leading to a higher nuclear accumulation which, in turn, has been shown to improve WUS protein stability[8,16].

## Results

**Exogenous CLAVATA3 prolongs WUSCHEL protein accumulation despite the rapid loss of expression.** To determine the

**WUSCHEL physically interacts with EXPORTIN proteins.** To further explore the role of CLV3-signaling in regulating the

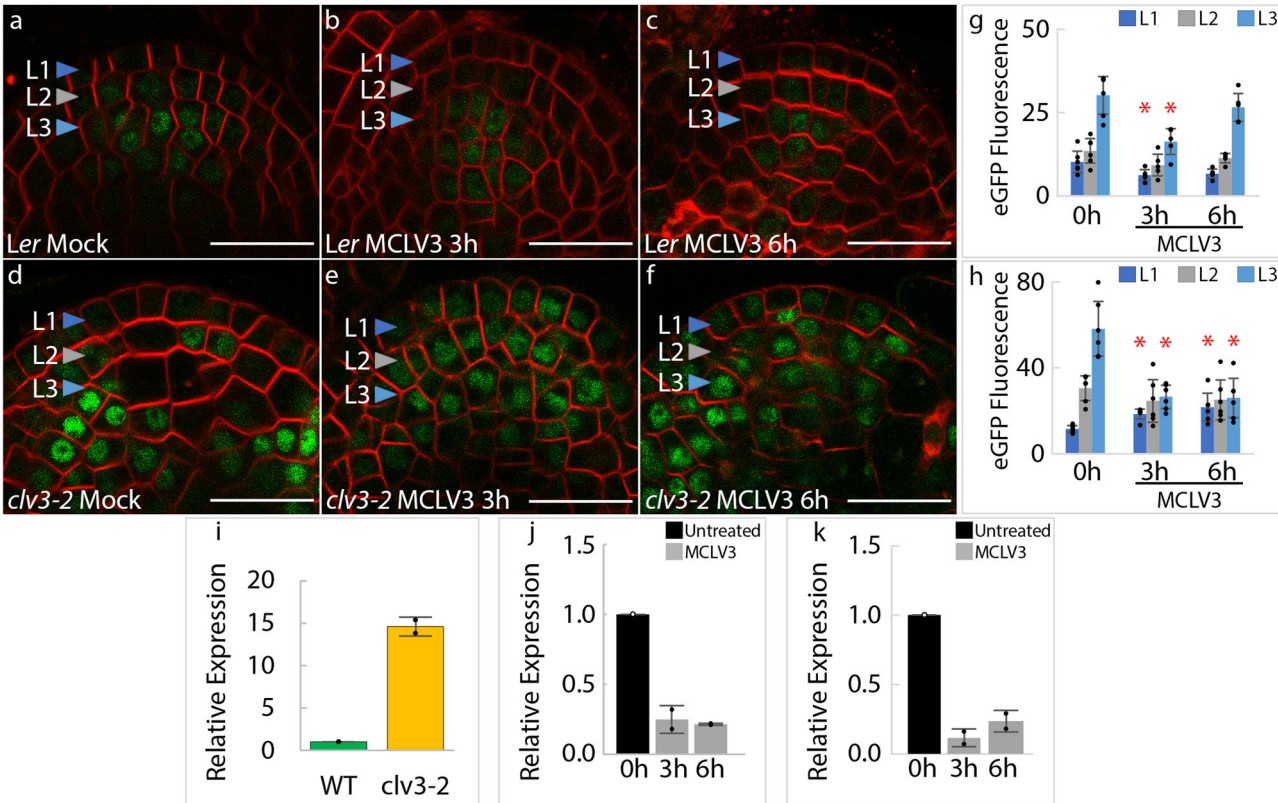

**Fig. 1 Short-term MCLV3 peptide application rapidly decreases *WUS* transcription but maintains the WUS protein.** *pWUS*::eGFP-WUS reporter response in 7-day old **a** mock-treated ($n = 6$), **b** 1 μM MCLV3 treatment for 3 h ($n = 5$), or **c** 1 μM MCLV3 treatment for 6 h ($n = 5$) in wild-type SAMs. *pWUS*::eGFP-WUS reporter response in 7-day old **d** mock-treated ($n = 6$), **e** 1 μM MCLV3 treatment for 3 h ($n = 6$), or **f** 1 μM MCLV3 treatment for 6 h ($n = 6$) in *clv3-2* SAMs. **g**, **h** Quantification of *pWUS*::eGFP-WUS fluorescence levels (mean ± s.d.) in **g** wild type or **h** *clv3-2* SAMs shown in **a–f**. The "*n*" represents the number of independently treated plants. The * represents a *p* value ≤0.05 when compared to WUS levels at 0 h in the same cell layer (Student's two-tailed *t*-test). eGFP (green) is overlaid on FM4-64 (red) plasma membrane stain. Scale bars = 20 μm. **i** Quantitative RT-PCR analysis showing the relative combined *WUS* and *eGFP-WUS* expression (mean ± s.d.) in 7-day-old wild type ($n = 2$) and *clv3-2* mutants ($n = 2$). The "*n*" represents the number of biological replicates. **j**, **k** Quantitative RT-PCR analysis showing relative combined *WUS* and *eGFP-WUS* expression (mean ± s.d.) in response to 1 μM MCLV3 treatment for 3 and 6 h in wild type (**j**) and *clv3-2* mutants (**k**). Two biological replicates were used for each time point. *UBQ10* was used as a reference gene to normalize the data and relative changes in *WUS* expression were determined relative to the levels at 0 h. Error bars represent the standard deviation (s.d.).

nuclear export of WUS, we tested whether WUS interacts with EXPORTIN proteins. There are two members of the EXPORTIN (XPO1) protein family in *Arabidopsis*: XPO1A and XPO1B, that function as receptors for nuclear export of nuclear-localized protein cargo[17]. Using Bimolecular Fluorescence Complementation (BiFC), we found a physical interaction between WUS with XPO1A and XPO1B in *N. benthamiana* leaf pavement cells (Supplementary Fig. 5). A previous structure-function analysis of the WUS protein revealed that the leucine-rich EAR-like [EARL] domain of WUS could function as a leucine-rich repeat NES because the point-mutations of the leucine residues increased nuclear WUS levels[8]. The WUS mutant protein carrying the same substitutions of the leucine residues failed to interact with XPO1A (Supplementary Fig. 5e, f) and XOP1B (Supplementary Fig. 5g, h) showing that the EAR-like domain of WUS interacts with EXPORTINS.

Curiously, we observed a higher accumulation of EAR-like domain mutant protein (*p35S*::eGFP-WUS [EARLM]) in the cytoplasm of transfected leaves when compared to the wild-type protein (Supplementary Fig. 5o, p). This suggested the possibility that an EAR-like domain may also be required for destabilizing WUS in the cytoplasm which could not be revealed by using the *pWUS*::eGFP-WUS [EARLM] since EAR-like domain also functions as an NES[8]. Therefore, we utilized *p35S*::eGFP-WUS-

[EARLM]-GR expressing SAMs to quantify the cytoplasmic levels of the mutant protein which revealed a significantly higher accumulation than the wild-type protein suggesting that EAR-like domain may have a second function in destabilizing the WUS in the cytoplasm (Supplementary Fig. 6). Such a coupled behavior where nuclear export signal also functions as a degron in the cytoplasm has been observed for Aryl hydrocarbon receptor, a ligand (2,3,7,8- tetrachlorodibenzo-*p*-dioxin (TCDD)) activated nuclear TF-AHR[18].

To explore the role of nuclear export regulation on the WUS protein gradient, we perturbed nuclear export using Leptomycin-B (LEP-B), an inhibitor of EXPORTIN function[19]. To test the efficacy of LEP-B treatment in blocking nuclear export, we first treated the *pWUS*::NES-eGFP-WUS, a WUS protein form carrying an exogenous NES derived from Rev protein of HIV-1 expressed from the *WUS* promoter[8]. *pWUS*::NES-eGFP-WUS protein predominantly accumulated outside the nucleus (Supplementary Fig. 7a–f), and the LEP-B treatment resulted in the nuclear accumulation showing the efficacy of LEP-B in inhibiting nuclear export in the SAM (Supplementary Fig. 7g–l).

**WUS nuclear export regulation occurs downstream of CLV3-signaling.** We then monitored the effects of LEP-B treatment for

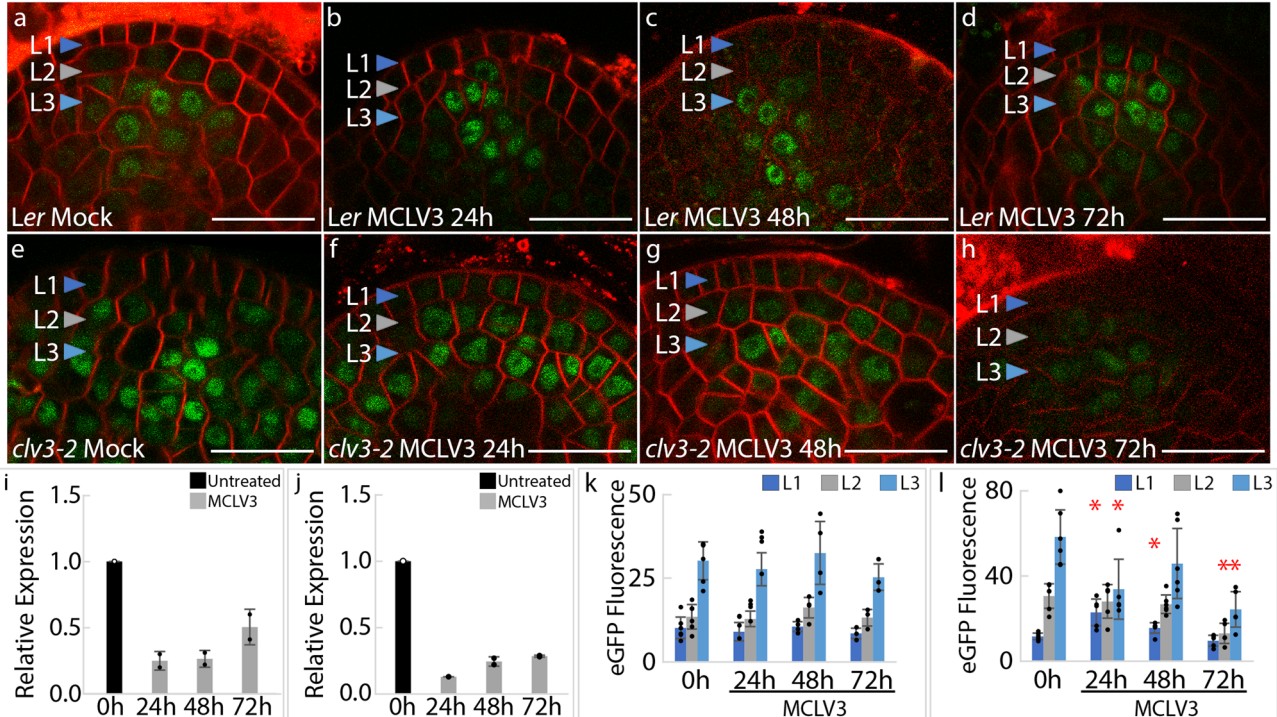

**Fig. 2 Long-term MCLV3 peptide application maintains the WUS protein long after the decrease in *WUS* transcription.** *pWUS::*eGFP-WUS reporter response in 7-day old **a** mock-treated (*n* = 6), 1 μM MCLV3 treatment for **b** 24 (*n* = 4), **c** 48 (*n* = 4), or **d** 72 h (*n* = 3) in wild-type SAMs. *pWUS::*eGFP-WUS reporter response in 7-day old **e** untreated (*n* = 6), 1 μM MCLV3 treatment for **f** 24 (*n* = 5), **g** 48 (*n* = 6), or **h** 72 h (*n* = 5) in *clv3-2* SAMs. In (**a–h**), The "*n*" represents the number of independently treated plants. (**i, j**) Quantitative RT-PCR was performed to determine the relative combined *WUS* and *eGFP-WUS* expression (mean ± s.d.) in 7-day old **i** wild type and **j** *clv3-2* mutants in response to 1 μM MCLV3 treatment for 24, 48, and 72 h. Two biological replicates were used for each time point. *UBQ10* was used as a reference gene to normalize the data and relative changes in *WUS* expression were determined relative to the levels at 0 h. **k, l** Quantification of *pWUS::*eGFP-WUS fluorescence levels (mean ± s.d.) in **k** wild type or **l** *clv3-2* SAMs from experiments shown in **a–h**. The * represents a *p* value ≤ 0.05 when compared to WUS levels at 0 h in the same cell layer (Student's two-tailed *t*-test). Error bars represent the standard deviation (s.d.). eGFP (green) is overlaid on FM4-64 (red) plasma membrane stain. Scale bars = 20 μM.

24 h on the *pWUS::*eGFP-WUS reporter in wild-type SAMs which revealed an increase in nuclear levels across all three cell layers though significantly higher nuclear levels were observed in the L3 (Fig. 4a–c). The regulation of nuclear export influences the cytoplasmic levels and diffusion of WUS into neighboring cells. Therefore, the steady-state estimation of WUS accumulation upon LEP-B treatment represents an output of a complex regulation involving all these processes which might explain the variable accumulation of WUS in the L1 and L2 layers. Therefore, to better assess the effects of LEP-B, we used Dex-inducible system-*p35S::*eGFP-WUS-GR that can transiently translocate into the nuclei of cells in all layers, upon Dex treatment. The 24 h Dex treatment alone leads to a pronounced destabilization of eGFP-WUS-GR in outer cell layers as discussed earlier (Fig. 3a–c). However, the co-treatment of Dex and LEP-B led to a stable nuclear accumulation of eGFP-WUS-GR (Fig. 3e) similar to the co-treatment of Dex and MCLV3 observed previously (Fig. 3d). These results show that blocking nuclear export leads to a stable nuclear accumulation of WUS.

Next, we tested whether LEP-B can block nuclear export in *clv3-2* mutants. The 24 h treatment of *pWUS::*eGFP-WUS:*clv3-2* plants with LEP-B revealed a significantly higher WUS protein accumulation in the L1 and L2 layers (Fig. 4d-f). Moreover, the co-treatment of *p35S::*eGFP-WUS-GR:*clv3-2* plants with Dex and LEP-B also led to a stable nuclear accumulation of eGFP-WUS-GR (Fig. 3j). The LEP-B-mediated inhibition of nuclear export in *clv3-2* mutants suggests that the nuclear export regulation could occur downstream of CLV3-mediated signaling. Similar nuclear enrichment observed upon MCLV3 treatment

also suggests that CLV3-signaling may offset nuclear export presumably through WUS protein modifications.

If CLV3-signaling offsets nuclear export alone, we expected a higher WUS protein accumulation in the cytoplasm of *clv3-2* mutants, however, which was not the case (Supplementary Fig. 8g), suggesting that CLV3-signaling may also be required for either stabilizing the WUS protein in the cytoplasm or inhibiting its diffusion into adjacent cells. To test the former possibility, we challenged the cytoplasmically-localized WUS (*p35S::*eGFP-WUS-GR) with exogenous application of MCLV3 (Supplementary Fig. 9). Our results show that MCLV3 treatment did not change the cytoplasmic WUS protein levels in both wild type (Supplementary Fig. 9a, b, e) and *clv3-2* (Supplementary Fig. 9c, d, f), showing that CLV3-signaling is not required for cytoplasmic stabilization of WUS and therefore may inhibit the diffusion of cytoplasmically-localized WUS protein into adjacent cells.

**Development and calibration of the computational model.** To understand the dual role of CLV3-signaling in inhibiting transcription of *WUS* and thereby its synthesis and the post-translational regulation offsetting the nuclear export of the WUS protein, we developed a computational cell-based model in three dimensions, based on the previous framework[5], to represent the individual cells of the SAM with the following assumptions: (a) WUS activates and represses *CLV3* expression in a concentration-dependent manner[9]; (b) CLV3 represses *WUS* transcription[10–12] and affects the N-C partitioning of WUS by inhibiting nuclear export as shown in this study; (c) WUS protein synthesis occurs in the cytoplasm and can translocate into the nucleus or diffuse into neighboring cells. WUS

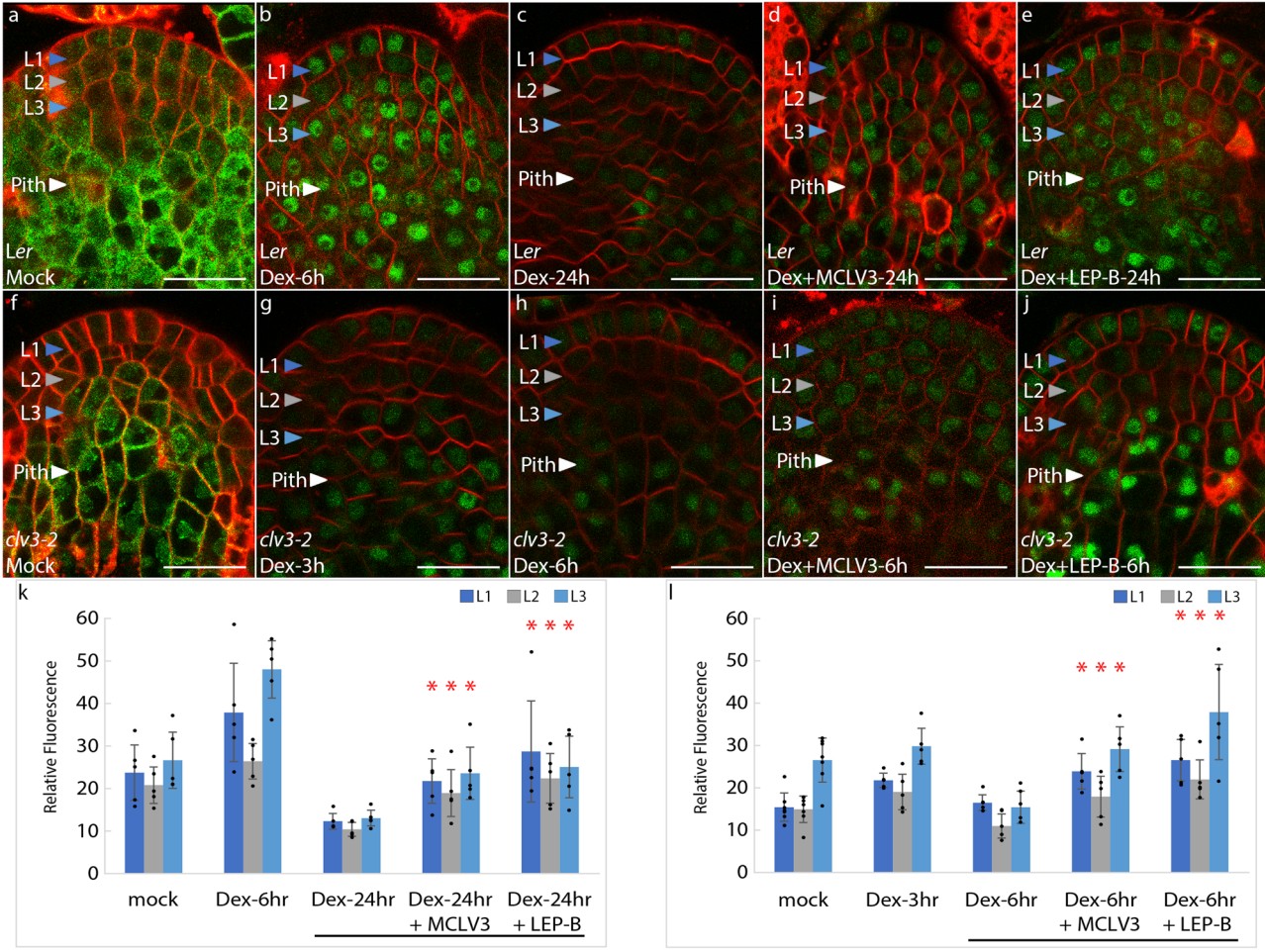

**Fig. 3 Nuclear stabilization of transiently translocated eGFP-WUS-GR protein upon MCLV3 peptide and Leptomycin-B treatments.** *p35S::*eGFP-WUS-GR reporter response in 7-day old **a** mock-treated (*n* = 5), **b** short-term [6 h] Dex-induction (*n* = 5), **c** long-term [24 h] Dex-induction (*n* = 5), **d** long-term Dex-induction [24 h] co-treated with 1 µM MCLV3 (*n* = 5), and **e** long-term Dex-induction [24 h] co-treated with 20 nM LEP-B (*n* = 5) in wild-type SAMs. *p35S::*eGFP-WUS-GR reporter response in 7-day old **f** mock-treated (*n* = 7), **g** short-term [3 h] Dex-induction (*n* = 5), **h** long-term [6 h] Dex-induction (*n* = 5), **i** long-term Dex-induction [6 h] co-treated with 1 µM MCLV3 (*n* = 5), and **j** long-term Dex-induction [6 h] co-treated with 20 nM LEP-B (*n* = 5) in *clv3-2* SAMs. The "*n*" represents the number of independently treated plants. **k**, **l** Quantification of *p35S::*eGFP-WUS-GR fluorescence levels (mean ± s.d.) in **k** wild type or **l** *clv3-2* SAMs from experiments shown in **a–j**. The * represents a *p* value ≤ 0.05 when compared to WUS levels from corresponding Dex treatments (Dex-24 h for wild type and Dex-6 h for *clv3-2*) from the same cell layer (Student's two-tailed *t*-test). Error bars represent the standard deviation (s.d.). eGFP (green) is overlaid on FM4-64 (red) plasma membrane stain. Scale bars = 20 µM.

in the nucleus and cytoplasm were modeled as two different variables to account for N-C partitioning; (d) cytokinin-signaling in the RM acts on the WUS-box motif which also functions as the nuclear retention signal (NRS) to increase nuclear WUS levels[8,16]; (e) the EAR-like motif of WUS functions as the NES[8], and (f) WUS can self-stabilize in the nucleus upon reaching a threshold[16]. A diagram of the signaling model influencing the subcellular distribution and diffusion of the WUS protein is shown in Fig. 5a and a detailed description of the modeling framework is provided in the supplementary materials (Supplementary Fig. 10). The WUS gradient was calibrated by comparing experimentally quantified N-C distribution of the WUS protein from individual cell layers of the wild type (Supplementary Fig. 8h, j) and *clv3-2* (Supplementary Fig. 8i, j). Furthermore, the nuclear export and retention functions were calibrated by comparing with WUS protein accumulation in EAR-like [EARLM] (Supplementary Fig. 11d) and WUS-Box mutants [WBM], respectively (Supplementary Fig. 11c). The calibrated model was able to attain the experimentally observed ratio of the nuclear WUS levels between the L3 and the L1 layers in the wild type and the *clv3-2* mutants (Fig. 5b–d).

**The dual transcriptional and post-translational regulation provides robustness at higher WUS synthesis.** With the calibrated model, we probed the significance of CLV3-mediated post-translational regulation of WUS by examining the effects of perturbations of the maximal *CLV3* or *WUS* expression rate on the spatial patterns of *CLV3* and *WUS* expression domains and the WUS protein gradient in simulations. In the model lacking the CLV3 regulation, the WUS protein gradient is extremely sensitive to the *WUS* expression level, WUS protein diffusion rate, nuclear export rate, and stability, as observed in the sensitivity analysis (Supplementary Fig. 12). The wild-type *CLV3* has been shown to express in the repressive window of the WUS concentration since the partial depletion of WUS initially leads to CLV3 over activation[9] (Supplementary Fig. 13). In the model lacking the post-translational regulation within the repression window of the WUS concentration, an increase in *WUS* expression rate led to downregulation of *CLV3* expression, and a decrease in *WUS* expression rate led to the expression of *CLV3* in inner cell layers (Fig. 6e and Supplementary Fig. 8k). In contrast, enabling the post-translational regulation was able to maintain *CLV3* expression in the outer cell layers for the *WUS* expression

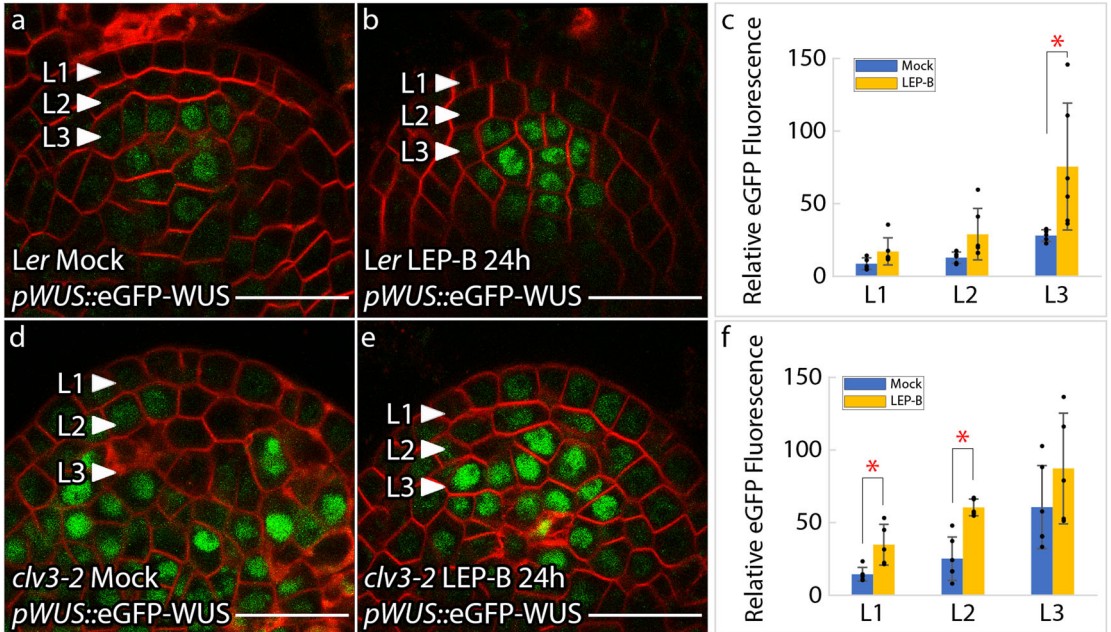

**Fig. 4 Stable nuclear accumulation of the WUS protein (*pWUS*::eGFP-WUS) in the L1 layer of *clv3-2* mutants upon Leptomycin-B treatments. a**, **b** Wild-type plants with *pWUS*::eGFP-WUS reporter in response to **a** mock (*n* = 5) or **b** 20 nM LEP-B (*n* = 6) treatments. **c** Quantification of *pWUS*::eGFP-WUS fluorescence levels (mean ± s.d.) from treatments shown in **a**, **b**. The "*n*" represents the number of independently treated plants. **d**, **e** *clv3-2* mutant plants with *pWUS*::eGFP-WUS reporter in response to **d** mock (*n* = 6) or **e** 20 nM LEP-B (*n* = 5) treatments for 24 h. **f** Quantification of *pWUS*:: eGFP-WUS fluorescence levels (mean ± s.d.) from treatments shown in **d**, **e**. The "*n*" represents the number of independently treated plants. The * represents a *p* value ≤ 0.05 when compared to WUS levels from mock treatments in the same cell layer (Student's two-tailed *t*-test). Error bars represent the standard deviation. eGFP (green) is overlaid on FM4-64 (Red) plasma membrane stain. Scale bars = 20 µM.

rate across the same range (Fig. 6f and Supplementary Fig. 8l). Our simulations also revealed that the *WUS* expression domain and the WUS protein gradient were close to the unperturbed condition with both the transcriptional and post-translational regulations operating simultaneously when compared to the transcriptional regulation alone, showing that the dual regulation provides robustness.

**The dual regulation along with the requirement of CLV3 production from the L1/outer cell layers provides robustness at lower WUSCHEL synthesis.** Conversely, a reduced *WUS* expression rate led to the internalized expression of *CLV3* in inner cell layers even with the dual regulation (Fig. 6f and Supplementary Fig. 8l), which is not consistent with the experimental observation where *CLV3* expression was maintained in the outer cell layers of the CZ upon partial depletion of WUS[9]. The extreme sensitivity of *CLV3* at lower WUS in the simulations was likely due to a dramatic inhibition of *WUS* expression. However, earlier experiments have revealed that the *WUS* expression domain was maintained in cytokinin receptor mutants[16] and in *lostmeristem (lom)/hairymeristem (ham)* mutants[20] despite the internalization of *CLV3* expression. Moreover, the *CLV3* promoter variants lacking the WUS binding *cis*-elements led to internalized *CLV3* expression but did not terminate shoot meristem growth[9]. On the contrary, the expression of *CLV3* from the L1 layer promoter has been shown to terminate shoot meristems in 100% of the transgenic plants screened[21]. Additionally, providing the bioactive CLV3 peptide suppressed *WUS* expression (Fig. 1j, k) and terminated shoot meristems[14]. Taken together, these observations reveal that *CLV3* expressed in the L1 or outer cell layers is far more effective in terminating shoot meristem growth possibly due to the localization of the ligand processing or modification machinery in these cells. We incorporated this additional mechanism in the model by decreasing the potency of *CLV3*

expressed in the inner cell layers. A 25% decrease in the efficacy of CLV3 generated in the inner cell layers along with the dual transcriptional and the post-translational regulation was able to maintain *CLV3* expression in the outer cell layers over the same range of reduced WUS expression rate (Fig. 6g and Supplementary Fig. 8m). Similar robustness was observed upon perturbing *CLV3* expression rates (Supplementary Fig. 14). The dual regulation was able to maintain *WUS* and *CLV3* expression and the WUS protein gradient at a lower *CLV3* expression rate compared with the transcriptional regulation only (Supplementary Fig. 14b, e). The dual regulation along with the spatial requirement of effective *CLV3* was able to maintain *CLV3* and *WUS* expression and the WUS protein gradient even at a higher *CLV3* expression rate compared with the model without the spatial requirement of *CLV3* (Supplementary Fig. 14c, f). Perturbation studies performed on different biologically relevant parameter sets showed that at lower *WUS* expression levels, the dual regulation resists the internalization of *CLV3* better than the transcriptional regulation only thereby maintaining the *WUS* transcription and WUS gradient (Supplementary Figs. 15, 16). At higher *WUS* expression levels, the dual model resists severe repression of *CLV3* thereby preventing the overproduction of WUS (Supplementary Figs. 16–18). The proposed mechanism improves robustness over different parameter sets, which hightlights its importance in maintaining the WUS gradient and stem cell homeostasis (see Methods section for more details; Supplementary Figs. 15–18).

**Feedback between subcellular partitioning of WUSCHEL and concentration-dependent transcriptional switching of CLA-VATA3 maintains homeostasis.** The simulations suggest that the CLV3-mediated dual control and the spatial requirement for the production of the effective CLV3, coupled with the concentration-dependent reversible regulation of *CLV3*

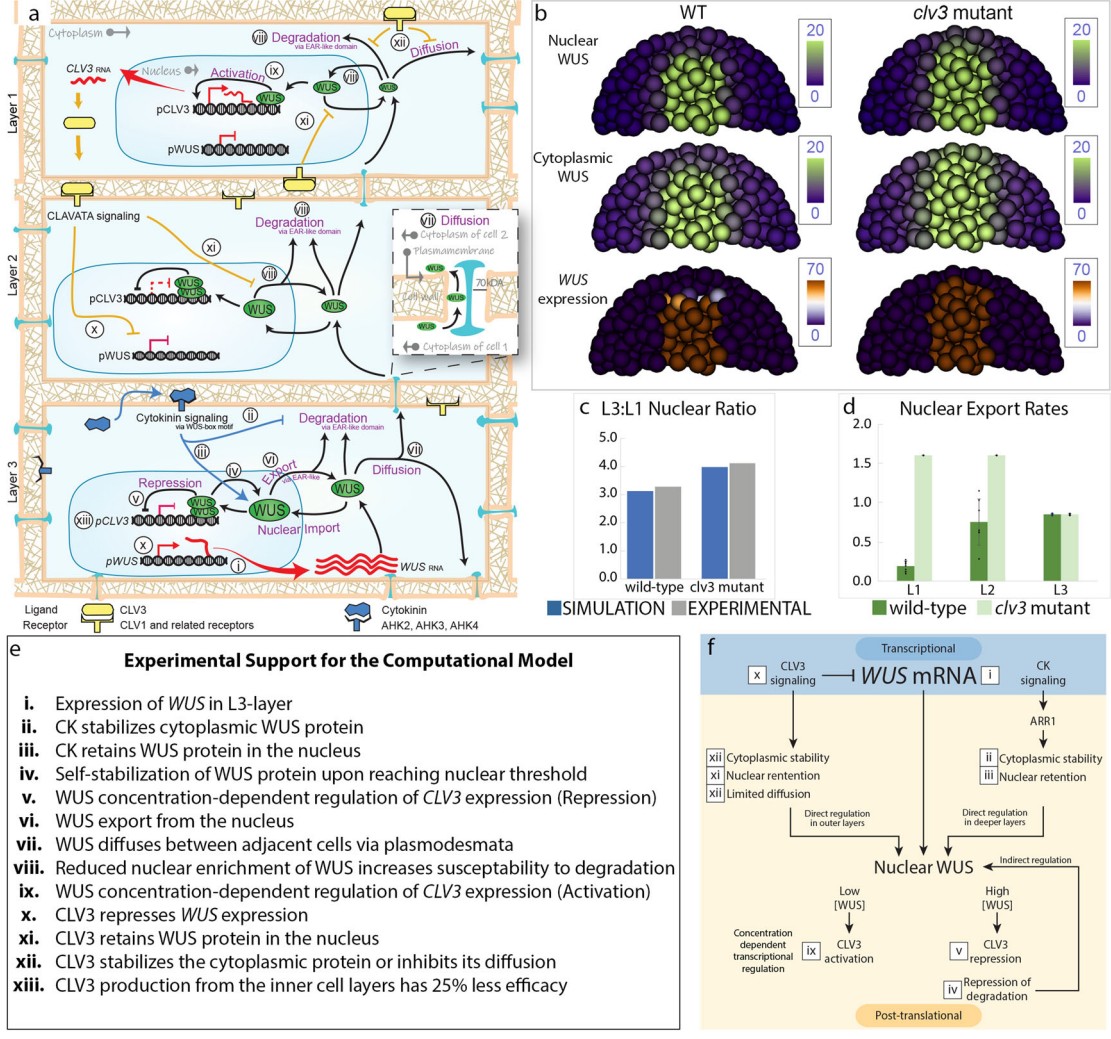

**Fig. 5 Multi-cell layer model for the regulation of the WUS protein gradient. a** The WUS-CLV3 feedback loop model with WUS-mediated concentration-dependent regulation of *CLV3* and CLV3-mediated dual transcriptional and post-translational regulation of WUS. **b** Model simulation of nuclear WUS protein, cytoplasmic WUS protein, and *WUS* expression in wild type and *clv3-2* mutant SAMs. **c** Ratio of L3 to L1 (L3:L1) nuclear WUS (mean ± s.d.) for wild type and *clv3-2* mutants. For experimental quantification independently imaged wild type ($n = 7$) and *clv3-2* ($n = 8$) SAMs were considered. For simulations "*n*" represents the number of cells in wild-type L1 [$n = 8$] and L3 [$n = 6$], and in *clv3-2* L1 [$n = 8$] and L3 [$n = 6$]). **d** Model prediction of nuclear export rates across cell layers (L1 [$n = 8$], L2 [$n = 7$], and L3 [$n = 6$], mean ± s.d.) of wild type and *clv3-2* mutants. The "*n*" represents the number of cells considered for quantification. **e** Transcriptional and post-translational experimental support in the computational model. **f** Simplified model of transcriptional and post-translational regulation of WUS.

transcription provides robustness to the WUS gradient (Fig. 5a). We earlier observed in wild-type SAMs, the WUS protein accumulation remained largely unchanged upon 24, 48, and 72 h of MCLV3 treatments (Fig. 2b–d) despite a severe downregulation in *WUS* transcript levels within 24 h of treatment (Fig. 2i). This unchanged overall WUS protein level could be due to a decrease in nuclear export rates observed in simulations (Fig. 5d and Supplementary Fig. 14d–f), which compensates for the reduction in *WUS* transcription. A similar treatment of the *clv3-2* mutants revealed a significant increase in WUS protein level in the L1 and a decrease in the deeper cell layers (Figs. 1e, f, h, and 2f, l) along with a severe reduction in *WUS* transcript level (Figs. 1k and 2j). A much more significant change in the WUS accumulation pattern in the L1 layer of *clv3* mutants could be due to the lack of WUS-mediated concentration-dependent regulation of the native *CLV3* promoter showing the importance of the feedback between WUS-concentration and the reversible regulation of *CLV3* levels in fine-tuning the WUS gradient. Finally, to test the importance of nuclear export-mediated control of nuclear concentration on the

*CLV3* expression, we examined the effect of LEP-B treatment, which has been shown to increase WUS level (Fig. 4a–f), on the *pCLV3*::H2B-mYFP expression. The LEP-B treatment for 24 h resulted in a reduction of *CLV3* expression in both wild type (Supplementary Fig. 19a–d) and *clv3-2* mutant SAMs (Supplementary Fig. 19e–h). Taken together, these results show the importance of the feedback between CLV3-mediated post-translational regulation of nuclear WUS level and the concentration-dependent activation-repression of *CLV3* expression in regulating the WUS protein gradient.

## Discussion

Our analysis shows extreme sensitivity of *CLV3* expression and in turn *WUS* expression to small perturbations in either of the regulators with the transcriptional regulation alone when *CLV3* is expressed within the repression window of WUS concentration. This is because a small increase in WUS levels represses *CLV3* expression resulting in an even higher synthesis of WUS (Fig. 6a). However, with the CLV3-mediated post-translational regulation,

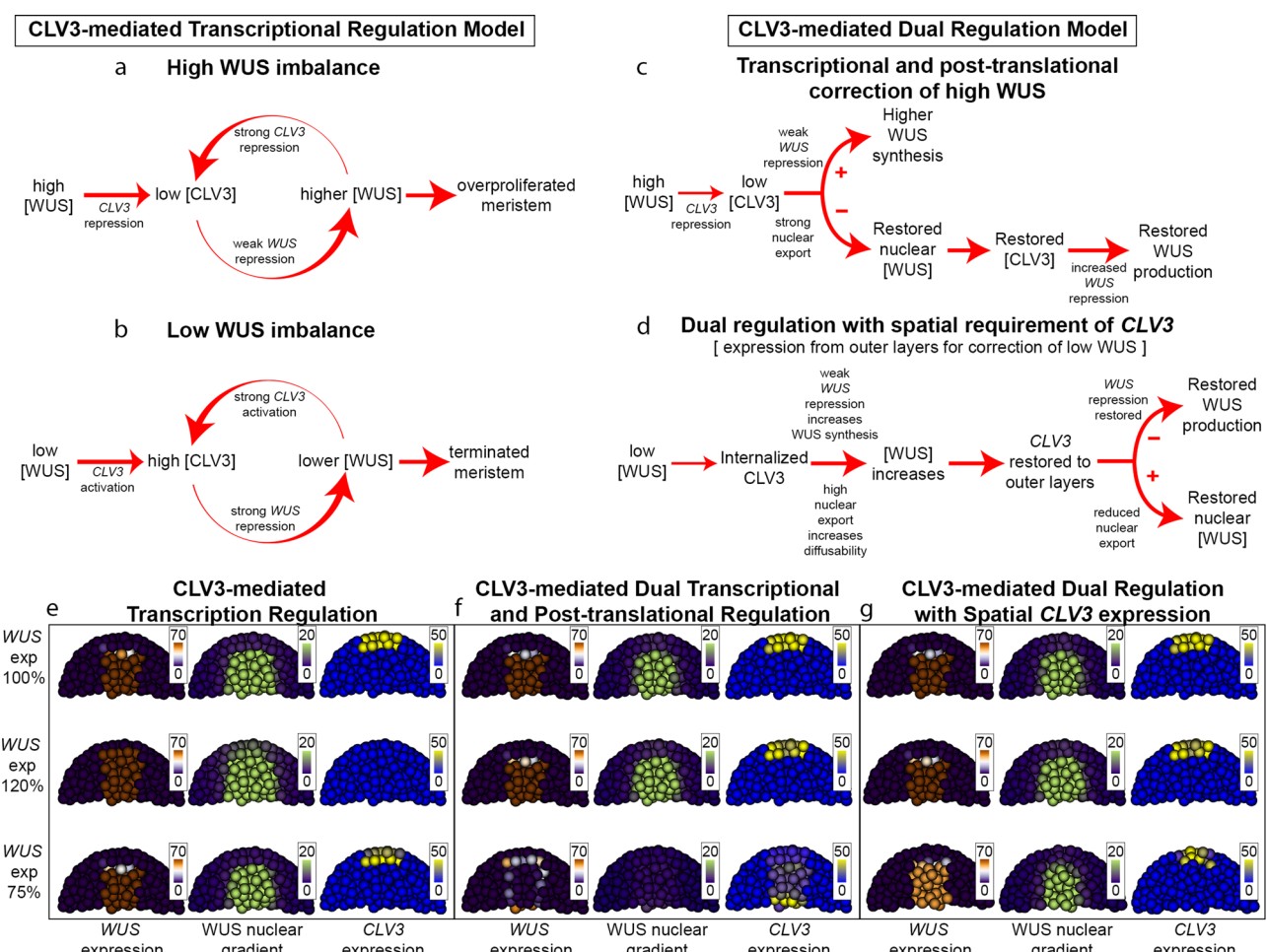

**Fig. 6 Dynamics of WUS-CLV3 feedback in simulations with CLV3-mediated transcriptional regulation, dual transcriptional and post-translational regulation, and dual regulation with the spatial requirement of *CLV3* expression from the outer layers, upon perturbing the *WUS* expression rate. a** In the transcriptional regulation model, high WUS concentration represses *CLV3*, which relieves *WUS* repression leading to more WUS production. **b** Low WUS concentration in the transcriptional regulation model activates *CLV3* which continues to suppress *WUS* expression, lowering WUS levels. **c** In contrast in the CLV3-mediated dual regulation model, high WUS leads to low *CLV3* which leads to increased nuclear export decreasing the nuclear concentration and increased WUS synthesis generating new WUS molecules. **d** Low WUS levels in the dual regulation model with the spatial requirement of *CLV3* expression from the outer cell layers leads to internalization of *CLV3* that is less potent in repressing *WUS* and reducing nuclear export which leads to increased synthesis of diffusible cytoplasmic WUS to restore WUS levels in the outer layers. With WUS levels restored, *CLV3* expression gets restored back to the outer layers which in turn reduces the nuclear export of WUS in the outer layers to restore WUS production and the nuclear WUS concentration. **e**–**g** *WUS* expression (brown), the WUS nuclear protein gradient (green), and *CLV3* expression (yellow) in the SAM upon perturbations in *WUS* expression level in simulated **e** CLV3-mediated transcriptional regulation model, **f** CLV3-mediated dual transcriptional and post-translational regulation model, and **g** CLV3-mediated dual regulation with the spatial requirement of *CLV3* expression from outer layers model.

an increase in WUS levels represses *CLV3*, which in turn promotes excessive nuclear export leading to a lower WUS level that reactivates *CLV3* (Fig. 6c). With the transcriptional regulation alone, reduction in WUS levels activates more *CLV3*, further lowering *WUS* transcription (Fig. 6b). However, with the CLV3-mediated post-translational regulation, a minor decrease in WUS levels leads to an increase in *CLV3* expression in the outer cell layers, which decreases the nuclear export of WUS to increase nuclear WUS levels. A drastic decrease in WUS could lead to the internalization of *CLV3*, which is less potent, leading to an increase in *WUS* transcription and an increase in nuclear export, thus restoring the outer cell layers with new WUS and reactivating *CLV3* (Fig. 6d). The combination of dual transcriptional and post-translational regulation along with the spatial regulation of *CLV3* can maintain the WUS nuclear gradient under fluctuating *WUS* expression levels, as verified by the computational analysis.

Our work provides an explanation for the tolerance of the SAM growth observed under a wide range of *CLV3* levels achieved through the expression from the native CLV3 promoter[22] which can balance *WUS* expression and N-C dynamics to maintain the WUS nuclear gradient. In contrast, the overexpression of *CLV3* from heterologous L1 layer promoter resulted in meristem termination[21] which could be due to the lack of concentration-dependent communication with the promoter. The competing models of *CLV3* regulation involving the HAM regulators have been published[23,24]. However, they do not provide an explanation for how a physical interaction between WUS and the rib meristem-localized HAM proteins leads to a downregulation of *CLV3* expression in the CZ. Understanding this non-cell-autonomous effect is required to integrate the HAM function into the models proposed here.

Our previous work shows that WUS protein stability/turnover depends on nuclear concentration, which is, in turn, regulated by

the N-C partitioning[8,9,16]. A relatively higher nuclear WUS concentration stabilizes the protein to repress *CLV3*. In contrast, a lower nuclear WUS destabilizes the protein to activate *CLV3* suggesting a need for balancing the de novo synthesis and turnover of WUS molecules to maintain *CLV3* expression. Our work showing the destabilization of the nuclear-exported WUS protein in the cytoplasm by using the EAR-like domain may create space for utilization of "new WUS species" that move into the nuclei of the CZ cells from the RM to orient *CLV3* transcription. Thus, the same CLV3 signal-regulating both the production of WUS and its subsequent N-C partitioning establishes a seamless connection between synthesis of new WUS molecules and replenishment of WUS to regulate nuclear concentration. The nuclear WUS concentration, in turn, regulates CLV3 levels through concentration-dependent transcriptional switching of *CLV3*, thus forming a self-regulating feedback system.

The N-C partitioning has been shown to influence WUS diffusion into adjacent cells[2,8]. Therefore, feedback between CLV3-mediated regulation of N-C partitioning of WUS and the WUS concentration-dependent regulation of *CLV3* also determines cytoplasmic levels that need to be regulated to control the rate of WUS movement into adjacent cells. The involvement of the EAR-like domain in both the nuclear export and subsequent degradation of WUS in the cytoplasm may better coordinate these processes in regulating the N-C ratio and WUS diffusion. Our earlier study has shown that a fraction of the plants expressing the EAR-like domain mutant WUS developed bigger SAMs[8] which could be attributed to the higher pool of cytoplasmically-stable WUS available for diffusion into the adjacent cells, reinforcing the importance of the EAR-like domain in the regulation of N-C ratios in controlling the size of the stem cell domain and the meristem size.

Our work favors the hypothesis that CLV3-signaling may inhibit lateral diffusion of WUS into adjacent cells. In our model the same CLV3 strength used for restricting lateral diffusion significantly blocked WUS movement into the outer cell layers suggesting that either CLV3 does not inhibit diffusion into the apical cell layers or requires lower strength than what is needed for inhibiting lateral diffusion. It is conceivable that such observed differences in simulations may in part be required for restricting lateral diffusion of WUS based on the following observations. The transient depletion of *CLV3* has been shown to result in a staggered centripetal expansion of CLV3 promoter activity leading to an increase in stem cell numbers and meristem size[25] which suggests that WUS could enter into the centrally-located cells in the L1 and L2 layers and then diffuses laterally. Moreover, the cells in the L1 and the L2 layers that divide into anticlinal planes may contain higher numbers of primary plasmodesmata (PDs) than the non-dividing cell walls that separate L3 from L2, and L2 from L1 layers. Perhaps these anatomical differences contribute to higher rates of WUS diffusion laterally than into the apical cell layers. Therefore, a signal such as CLV3 may be required for inhibiting lateral diffusion of WUS. Future work on the analysis of receptor systems involved in perceiving CLV3 signal, PD distribution in SAMs, and WUS protein modifications may provide insights into the regulation of WUS diffusion and stem cell maintenance.

## Methods

**Plant growth conditions and genotypes**. All plants were grown at 25 ºC under continuous light. For seedling imaging experiments, all plants were grown on ½ MS and then transferred to plates containing either 1 μM scrambled control of the CLV3 peptide (sCLV3) (GenScript), 1 μM CLV3 peptide (MCLV3) (GenScript), 20 nM Leptomycin-B (LEP-B) (Enzo), or 10 μM Dexamethasone (Dex) (Sigma) for specified time-course and imaged at 7–8 days old. The WUS protein reporters: *pWUS*::eGFP-WUS[2], *pWUS*::NES-eGFP-WUS[8], *p35S*::eGFP-WUS-GR[8], and *p35S*::eGFP-WUS-[EARLM]-GR[8]; the control reporter: *p35S*::eGFP-GR[8]; the *CLV3*

reporter: *pCLV3*::H2B-myFP[9]; the cytokinin-signaling reporter: *pTCS*::mGFP-ER[26]; and the null mutant: *clv3-2*[11] have been previously described. The peptide sequences for sCLV3 and MCLV3 peptides are listed in Supplementary Table 4.

**Sample preparation and confocal microscopy**. To prepare seedlings for imaging experiments, seedlings were embedded in 4% agarose warmed to 60 ºC to generate a block for tissue sectioning. Excess agarose around the seedlings was trimmed using razors, tweezers, and a Zeiss Stemi 2000-C dissecting microscope to orient the seedlings into a vertical position. Seedlings were then sliced into two halves using a feather polished razor blade, immediately transferred to 3%-FM4-64x plasma membrane staining solution for 5–10 min, and then placed on a slide with a coverslip containing ddH$_2$O maintained on ice until ready for imaging. Sectioned seedlings were imaged with a 40x objective lens on a Leica SP5 Inverted Confocal microscope. We determine the median section of the SAM by identifying the region of the SAM where the center apex reaches its highest peak, which we assume to be the median section (see Supplementary Fig. 20). eGFP fluorescence was detected with 488 nm excitation and emission was collected between 500–550nm. FM4-64x membrane stain was detected using 543 nm excitation and emission was collected between 600–650nm. H2B-mYFP fluorescence was detected with 488 nm excitation and the emission was collected between 500–580nm.

For inflorescence meristems, meristems were cut from 3–4-week-old bolting plants and fixed in an imaging box containing 4% agarose. Flowers were dissected away from the meristem submerged in water and then allowed to dry before placing a drop of 3%-FM4-64x plasma staining solution for 30 min. Inflorescence meristems were imaged with a 40x dipping lens on a Zeiss LSM 880 Upright Confocal microscope. Detection of eGFP fluorescence and FM4-64x membrane stain was detected using 488 and 561 nm excitation, respectively. Emission was collected using the Airyscan detector. Following imaging, meristems were fixed (4% paraformaldehyde, 4% DMSO, 1xPBS) at room temperature for 30 min and stained with DAPI (ThermoFisher) for 120 min. DAPI does not penetrate the meristem well under our imaging conditions unless the tissue is fixed; however, fixing the tissue comprises the WUS fluorescence signal (see Supplementary Fig. 8a) so to overcome this, we first image the inflorescence meristem to capture the WUS fluorescence signal as a reference and align the data post-acquisition. DAPI was detected using 405 nm excitation and detected using the Airyscan detector. To align the images together, confocal micrographs of eGFP-WUS fluorescence with FM4-64x stain and DAPI were registered together with Photoshop and ImageJ using the nuclei and cell boundaries as landmarks to align WUS with the nuclear counterstain (see Supplementary Fig. 8b).

**Quantification of fluorescence levels from images**. To quantify WUS levels from inflorescence meristems, the slices of the WUS protein reporter, staining of cell boundary, and the stained nucleus were loaded into ImageJ and isolated for the eGFP, FM4-64, or DAPI channel. Only cells from the center apex (3–5 cells wide) of the SAM were considered when quantifying WUS levels. The DAPI nuclear counterstain was used as a reference to determine the ROI of nuclear WUS from the eGFP channel. Accumulation of WUS in the cytoplasm was determined using a polygon ROI within a cell while excluding the nucleus (Supplementary Fig. 8b).

To quantify the nuclear and cytoplasmic levels of WUS from seedlings, confocal micrographs of the WUS reporter were loaded into ImageJ and isolated for the eGFP or FM4-64 channel. Only cells from the center apex (3–5 cells wide) of the SAM were considered when quantifying WUS levels. The nucleus was marked with an ellipsoid ROI and the cytoplasm was marked with a polygon within the cell which also excluded the nuclear domain (Supplementary Fig. 8c). WUS levels from cells for each cell layer were pooled and averaged to determine nuclear and cytoplasmic levels in each cell layer (Supplementary Fig. 8f, g). The averaged levels across layers were then used to calculate the N-C ratio (Supplementary Fig. 8j) and the L3:L1 nuclear ratio (Fig. 5c).

We compared the nuclear and cytoplasmic WUS levels between the inflorescence and vegetative meristems and found them to be similar (Supplementary Fig. 8d, e), which provides confidence in our estimation of N-C levels.

To measure changes in WUS nuclear levels from chemical treatments, only the nuclear boundary was marked with an ellipsoid, and levels from cells for each cell layer were pooled and averaged to determine chemical treatment effects on WUS levels in each cell layer. To measure changes in *CLV3* expression levels and the change in a number of *CLV3* positive cells from LEP-B treatment, confocal micrographs of *CLV3* promoter activity were loaded into ImageJ and isolated for the YFP or FM4-64 channel. Only 3–5 center apex cells of the SAM were considered when measuring *CLV3* expression levels, whereas all cells of the SAM were considered when counting the number of *CLV3* positive cells. *CLV3* levels were determined by marking the boundary of the nuclear reporter with an ellipsoid and levels from cells for each cell layer were pooled and averaged to determine LEP-B effects on *CLV3* levels in each cell layer. The effect of LEP-B on the number of *CLV3* positive cells was determined by averaging the number of YFP positive cells from each cell layer. Statistical analysis comparing levels of a signal for each cell layer between treatments was performed using Student's two-tailed *t*-test with a *p* value ≤ 0.05.

**Quantitative RT-PCR**. Two biological replicates of tissue from 7-day-old L*er* or *clv3-2* seedlings containing the *pWUS::eGFP-WUS* reporter from the CLV3 peptide (MCLV3) or scrambled peptide control (sCLV3) application for the specified time-course was flash frozen and ground in liquid nitrogen. RNA was extracted according to the GeneJet Plant RNA Purification kit (Thermo Scientific). cDNA synthesis was performed using M-MuLV Reverse Transcriptase (New England Biolabs). Luna SYBR reagents (NEB) were used to perform quantitative RT-PCR of the combined expression of the endogenous WUS and eGFP-WUS reporter on the BioRad quantitative thermocycler. The primers used for PCR amplification are listed in Supplementary Table 5. The analysis was performed for duplicate samples and the quantification was normalized to the *UBQ10* gene.

**Bimolecular fluorescence complementation**. BiFC analysis was performed in leaf cells of 3–4-week-old *N. benthamiana* as described in earlier studies[8]. Coding sequences of *XPO1a* and *XPO1b* were first cloned into pCR BLUNT II (Thermo Scientific). DNA fragments were then digested and fused to the full-length eGFP (aa1-241), the N-terminal fragment of eGFP (N-eGFP; aa1-aa155), or the C-terminal fragment of eGFP (C-eGFP; aa156-aa241) and then cloned into pENTR. LR recombination into the pMDC32 destination vector was then performed. Generation of full-length eGFP, N-terminal eGFP, and C-terminal eGFP fused WUS in pMDC32 destination vector was previously described[8]. Mutations for EAR-like domain were introduced with primers from Supplementary Table 5. Empty N-terminal eGFP and C-terminal GFP were used as negative controls. *Agrobacterium tumefaciens* GV3101 was transformed with the BiFC constructs and infiltrated into leaves of 3–4-week-old *N. benthamiana* plants using agroinfiltration buffer solution (10 mM MES pH 5.7, 10 mM MgCl₂, 20 uM Acetosyringone). Infiltration of corresponding BiFC plasmids along with the p19 suppressor of gene silencing plasmid was introduced as a 1:1:1 cocktail. Fluorescence of eGFP was monitored using a Leica SP5 Confocal microscope. eGFP fluorescence was detected with 488 nm excitation and emission was collected between 500–525nm. Plastid autofluorescence emission was collected between 631–700 nm.

**Description of the computational model**. We developed a cell-based model in three dimensions to model the WUS gradient based on a previous framework[5]. Each individual cell was represented by a ball and the SAM tissue consists of 1366 cells. The cell growth and cell division were ignored in this model because the biochemical signaling network reaches the steady-state at a much faster time scale compared with the rates of cell division. Subcellular partitioning of WUS protein was considered by introducing two variables, $W_n$ for the WUS protein in the nucleus and $W_c$ for the WUS protein in the cytoplasm, into the model. Specifically, WUS protein synthesized in the cytoplasm can either diffuse into neighboring cells or be imported into the nucleus to bind DNA to activate *CLV3*. Specific signals were included to regulate nuclear export, protein stability, and diffusion of WUS, shown in the sketch provided in Fig. 5a, and reaction-diffusion equations were applied to model the dynamics. More modeling details were provided in the following and Supplementary Fig. 10.

The variable $W$ denotes the *WUS* mRNA level with a production term and degradation term in the equation. Its production $W_p(x, y, z)$ is constant in the central region of the SAM including the L2 and a few cell layers in the lateral direction within RM due to inhibition from cell layers beneath by hypothetical differentiation signals which are not included in this model. *WUS* transcription is repressed by *CLV3*, which is modeled by a Hill function. In the equation of $W_c$, it involves the diffusion term, which is approximated by passive transport[5], i.e., summation of fluxes between a given cell and its neighboring cells as shown below,

$$D_w \cdot \sum_{\text{cell } j \text{ and } i \text{ are neighbors}} \frac{A_{ij}}{l_{ij}}([W_c]_j - [W_c]_i),$$

where $D_w$ is the diffusion coefficient, $A_{ij}$ is the contacting area between two neighboring cells, $l_{ij}$ is the distance between the centers of two neighboring cells. If $D_w$ is not a constant and is affected by some signal, the average concentration between neighboring cells is used to regulate the diffusion coefficient. The synthesis term of $W_c$ is linearly dependent on *WUS* mRNA level, and the degradation term was decreased by CK signal based upon the fact that protein stability can be improved by CK[16]. CK and CLV3 signals reduce nuclear export because of their action on nuclear retention and inhibition of nuclear export, respectively. The nuclear import was assumed to have a constant rate due to the fact that structure-function analysis of the WUS protein did not reveal a clear nuclear import signal[8]. The equation of $W_n$ involves the degradation term which is reduced by $W_n$ itself, based on the experimental data on the concentration-dependent self-stabilization of the WUS protein in the nuclei[16]. The equation of $W_n$ has the same terms modeling nuclear import and export as those in the equation of $W_c$. CLV3 protein dynamics is modeled by a reaction-diffusion equation with the production term as a hat shape function of $W_n$, corresponding to the concentration-dependent activation of *CLV3*, diffusion with a constant coefficient, and degradation at a constant rate. CK signaling is modeled by including cytokinin ligand, receptor, and the ligand–receptor complex to model the binding and unbinding dynamics. The ligand is produced in deeper cell layers and can diffuse into every cell layer. The receptor is produced in deeper cell layers only. In the current model, CK signaling affects WUS protein stability and feedback from WUS to the CK signaling was not

considered because the *wus* null mutants maintain normal cytokinin response[16]. Therefore, all parameters involved in the CK equations are constants, not affected by other components in the model. The computational model was implemented in MATLAB by using the total variation diminishing Runge Kutta second-order method as the time integrator. The time step was set to be 0.01 to satisfy the stability and the final time was set to be 300 to guarantee the steady-state in each simulation.

**Calibration of the computational model**. The computational model developed in this study involved about 40 parameters. Although some of them were obtained based on a previous study[5] and some experimental observations directly, the new model introduced new components and additional feedback regulations for investigation and most of the parameters involved were not calibrated yet. It was always challenging to determine appropriate values for multiple interactive parameters required in a model simultaneously.

**Basic PDE model with reduced complexity**. In this work, we implemented the quasi-Monte Carlo simulations with Sobol sampling over a sufficiently large parameter space for a simpler model first. This simpler model did not have the extrinsic signals-CLV3 and CK, and only contained the framework describing the dynamics of WUS mRNA and protein in nucleus and cytoplasm at a cell-based structure. We reduced the number of parameters for the simpler model as below:

Considering the CK response and signaling are independent of the CLV3 and WUS levels, the CK signaling was included only for benchmarking to choose appropriate parameter values that can produce the cytokinin mutant behavior and those values were fixed in this study. This reduced the parameter number to 25. Subsequently, the global sensitivity analysis was applied to the *clv3* null mutant model by excluding both transcriptional (two parameters) and post-translational regulation of CLV3 (four parameters), as well as the *CLV3* transcription (four parameters) and diffusion (one parameter) and degradation (one parameter) of CLV3 peptides. The concentration-dependent self-stabilization of nuclear WUS was excluded in the global sensitivity analysis (four parameters) since WUS protein can also be stabilized by improving the nuclear import which does not depend on CLV3-signaling. The nuclear import rate of WUS protein was fixed because the biological experiments suggest nuclear import is not regulated by CLV3 as shown in the study and CK[16], and also no clear nuclear localization or import signal was detected in the WUS protein[8]. The parameters related to WUS synthesis domain were determined and fixed based on experimental images (two parameters). Therefore, the remaining five critical parameters investigated in the global sensitivity analysis were the synthesis rate of WUS mRNA ($A_1$), diffusion of WUS protein ($D_w$), the nuclear export rate of the WUS protein ($r_{ex}$), degradation rates of nuclear ($d_{wn}$) and cytoplasmic ($d_{wc}$) WUS, as shown in the following equations:

$$\frac{\partial [W]}{\partial t} = W_p(x, y, z) - d_W[W] \tag{1}$$

$$\frac{\partial [W_c]}{\partial t} = D_w \Delta [W_c] + r_c[W] - d_{wc}[W_c] + r_{ex}[W_n] - r_{im}[W_c] \tag{2}$$

$$\frac{\partial [W_n]}{\partial t} = -d_{Wn}\left(d_{\min} + \frac{d_{\max} - d_{\min}}{1 + \left(\frac{[W_n]}{k_{ww}}\right)^n}\right)[W_n] - r_{ex}[W_n] + r_{im}[W_c] \tag{3}$$

$$W_p(x, y, z) = A_1 \cdot I_{\{x^2 + y^2 \leq r_w^2, z \leq L_w\}}(x, y, z) \tag{4}$$

These parameters were chosen as the inputs of the calibration because they were the parameters describing the main physical processes and also the main targets of CLV3 and CK regulations. Here are the ranges for those parameters:
*WUS* mRNA synthesis rate: [0.1, 100]
Diffusion rate of WUS protein: [0.01, 10]
Degradation rate of WUS nuclear protein: [0.01, 10]
Degradation rate of WUS cytoplasmic protein: [0.01, 10]
Nuclear export rate of WUS protein: [0.01, 10]
We ran 220 simulations for two different random samples as well as the mixed parameter sets by exchanging one parameter values between two samples as required for calculating the sensitivity indexes following[27]. More specifically, two random samples $M_1$ and $M_2$ were generated with the same sample size following Sobol sequencing independently from the same ranges by using the MATLAB function *sobolset*:

$$M_1 = \begin{pmatrix} \vdots & x_1 & x_2 & x_3 & x_4 & x_5 & \vdots \end{pmatrix}, M_2 = \begin{pmatrix} \vdots & x'_1 & x'_2 & x'_3 & x'_4 & x'_5 & \vdots \end{pmatrix} \tag{5}$$

The total effect indices were defined as

$$\hat{S}_j^i = 1 - \frac{\hat{U}_{-j}^i - \hat{E}^2(y_i)}{\hat{V}(y_i)} \tag{6}$$

where $y_i$ was the $i$-th output, $n$ was the sample size, $j$ denoted the $j$-th input, and

$$\hat{E}^2(y_i) = \frac{1}{n}\sum_{r=1}^{n} y_i(x_{r1}, x_{r2}, x_{r3}, x_{r4}, x_{r5}) y_i(x'_{r1}, x'_{r2}, x'_{r3}, x'_{r4}, x'_{r5}) \quad (7)$$

$$\hat{U}_{-j}^{i} = \frac{1}{n-1}\sum_{r=1}^{n} y_i(x_{r1}, x_{r2}, x_{r3}, x_{r4}, x_{r5}) y_i(x_{r1}, \ldots, x'_{rj}, \ldots, x_{r5}) \quad (8)$$

$$\hat{V}(y_i) = \frac{1}{n}\sum_{r=1}^{n} y_i^2(x_{r1}, x_{r2}, x_{r3}, x_{r4}, x_{r5}) - \left[\frac{1}{n}\sum_{r=1}^{n} y_i(x_{r1}, x_{r2}, x_{r3}, x_{r4}, x_{r5})\right]^2 \quad (9)$$

Here the index $S_j^i$ represented the total effect of the $j$-th input on the $i$-th output. In this analysis, we computed the total effect indices of all five inputs on the WUS nuclear levels at L1, L2, and L3, respectively. The global sensitivity analysis showed that the WUS nuclear level in each of the three layers had a positive correlation with the WUS mRNA synthesis rate and a negative correlation with all the other parameters. This was shown to be biologically relevant in Supplementary Fig. 21a.

By using the quasi-Monte Carlo sensitivity analysis to investigate properties of the basic PDE model which involved much fewer parameters, we were able to select proper parameter sets that could generate results consistent with the experimental data. In particular, we concluded that the appropriate criteria to select the parameter values were as follows: WUS nuclear protein level in L3 was about threefold that in L1 and the ratio between WUS protein concentrations in nucleus and cytoplasm was close to 1 across different layers.

**Steady states of the ODE model.** Next, we calibrated the parameters associated with WUS-CLV3 feedback regulations by considering the following ordinary differential equation system and its steady states, which was an approximation of the steady-state obtained in the original PDE model by ignoring the spatial distribution:

$$\frac{d[W]}{dt} = \frac{W_p}{1 + \left(\frac{[CLV3]}{k_{cw}^1}\right)^{n_1}} - d_W[W] \quad (10)$$

$$\frac{d[W_c]}{dt} = r_c[W] - d_{wc}[W_c] + \frac{r_{ex}}{\left(1 + \left(\frac{[CLV3]}{k_{cw}^2}\right)^{n_2}\right)}[W_n] - r_{im}[W_c] \quad (11)$$

$$\frac{d[W_n]}{dt} = -d_{Wn}\left(d_{min} + \frac{d_{max} - d_{min}}{\left(1 + \left(\frac{[W_n]}{k_{ww}^1}\right)^{n_3}\right)}\right)[W_n] - \frac{r_{ex}}{\left(1 + \left(\frac{[CLV3]}{k_{cw}^2}\right)^{n_2}\right)}[W_n] + r_{im}[W_c] \quad (12)$$

$$\frac{d[CLV3]}{dt} = \frac{C_p}{\left(1 + \left(\frac{[W_n]}{k_{wc}^1}\right)^{n_4}\right)\left(1 + \left(\frac{[W_n]}{k_{wc}^2}\right)^{-n_5}\right)} - d_c[CLV3] \quad (13)$$

The system could be simplified into one nonlinear equation and solved using some numerical method. More specifically, the steady-state solutions were obtained by solving the following equations:

$$[CLV3]_{ss} = \frac{C_p/d_c}{\left(1 + \left([W_n]_{ss}/k_{wc}^1\right)^{n_4}\right)\left(1 + \left([W_n]_{ss}/k_{wc}^2\right)^{-n_5}\right)} \quad (14)$$

$$[W]_{ss} = \frac{W_p/d_w}{1 + \left([CLV3]_{ss}/k_{cw}^1\right)^{n_1}} \quad (15)$$

$$[W_c]_{ss} = \frac{d_{Wn}}{r_{ex}}\left(d_{min} + \frac{d_{max} - d_{min}}{\left(1 + \left([W_n]_{ss}/k_{ww}\right)^{n_3}\right)}\right)[W_n]_{ss} + \frac{r_{ex}/r_{im}}{\left(1 + ([CLV3]_{ss}/k_{cw}^2)^{n_2}\right)}[W_n]_{ss} \quad (16)$$

$$r_c[W]_{ss} - d_{wc}[W_c]_{ss} + \frac{r_{ex}}{\left(1 + ([CLV3]_{ss}/k_{cW}^2)^{n_2}\right)}[W_n]_{ss} - r_{im}[W_c]_{ss} = 0 \quad (17)$$

and the last nonlinear equation was solved numerically in MATLAB by the function *fzero*. Notice that there existed a unique stable steady state for this ODE model such that $W_n$ and CLV3 coexisted. This allowed us to calibrate the parameters, especially the EC50 coefficients in Hill functions which modeled specific regulations, by doing a local perturbation of every single parameter on the stable steady-state (Supplementary Fig. 21b). We also investigated the model by removing the post-translational regulation of CLV3 on WUS protein. The number of steady states remained the same. However, the steady-state which had $W_n$ and CLV3 coexisting and was stable in the system with dual regulations became unstable with only transcriptional regulation. This suggested that the post-translational regulation helped the WUS-CLV3 system reach a stable equilibrium with robustness. The results are provided in Supplementary Fig. 21c, d.

A detailed explanation of the choice of individual parameters involved in concentration-dependent *CLV3* expression, and CLV3-mediated transcriptional and post-translational regulations of WUS gradient is provided below:

**Parameters involved in CLV3 expression: $W_\theta^1$, $W_\theta^2$, $W_\theta^3$.** The *CLV3* expression parameters were chosen based on the WUS nuclear levels in the L1 layer. First, we estimated the WUS nuclear level in the L1 layer for wild type. The ratio of WUS protein between wild type and *clv3* mutant measured in experiments, along with the WUS protein level in the L1 layer of *clv3* mutant obtained from global sensitivity analysis, was used to compute the WUS protein level in the L1 layer in wild type. Then the *CLV3* activation peak $W_\theta^2$ was chosen to be close to the estimated WUS protein level. $W_\theta^1$ determined the WUS concentration threshold below which *CLV3* expression reduces more than 50% (Supplementary Fig. 13c). $W_\theta^3$ determined the WUS concentration threshold above which CLV3 expression reduces more than 50% (Supplementary Fig. 13a). The experimental evidence shows that the L1 nuclear level of WUS is slightly repressive to *CLV3*[9], i.e., it falls at the right-hand side of the *CLV3* activation peak (Supplementary Fig. 13a–c).

**Parameters involved in CLV3 diffusion and degradation: $D_c$, $d_c$, $A_2$.** These parameters determined the decay length of CLV3 peptides, i.e., how deep CLV3 peptides could diffuse from the L1 and L2 layers. In particular, the decay length of a diffusive molecule was equal to $\sqrt{D_c/d_c}$. In the calibration, we chose them to satisfy CLV3 peptides reaching at least L3 layers. In addition, $d_c$ together with the CLV3 synthesis rate $A_2$ determined the maximal level that CLV3 could reach, which is $A_2/d_c$. Therefore, we fixed $D_c$ and $A_2$, and perturb $d_c$ only to study the effects of the CLV3 decay length and the maximal level on the WUS gradient.

**Parameters involved in CLV3 transcriptional and post-translational regulations: $k_{cw}^1$, $n_1$, $k_{cw}^2$, $n_6$, $k_{cw}^3$, $n_3$.** The simulation results would be sensitive to $k_{cw}^i$ for $i = 1, 2, 3$ and they were chosen appropriately such that CLV3 regulations would take effect in the model. In particular, $k_{cw}^i$ are EC50 values in the Hill functions modeling the CLV3 regulations and they were chosen to be less than the maximal level of CLV3 in order to generate an effective CLV3 regulation. The Hill coefficients $n_1$, $n_3$, $n_6$ determined how nonlinear the regulations were depending on the CLV3 concentration, i.e., larger coefficients gave rise to more switch-like regulations. The dual function of CLV3 in restricting WUS transcription to the L3 and in regulating the nuclear levels of WUS in the outer layers requires a switch-like regulation, therefore high Hill coefficients were chosen in the model. This high nonlinearity could be due to the spatially restricted receptors or intracellular signaling networks that mediate the dual regulation of CLV3 in producing different outputs in closely spaced domains.

In summary, by integrating the results of quasi-Monte Carlo sensitivity analysis and the steady-state analysis of the corresponding ODE system along with the properties of CLV3 transcription and regulations, we were able to calibrate the computational model and obtained an appropriate parameter set that could generate simulation results close to the experimental data based on the observations on the protein levels in both nucleus and cytoplasm at different layers, as shown in Fig. 5.

**Local sensitivity analysis of the computational model.** To better understand the roles of different biological processes affecting the WUS gradient more precisely, we performed a local sensitivity analysis on the following parameters of the computational model without the extrinsic signals-CLV3 and CK regulation near the parameter set obtained in the calibration: synthesis rate of WUS mRNA ($r_w$), the diffusion rate of WUS ($D_w$), the degradation rate of $W_n$ ($d_{wn}$) and $W_c$ ($d_{wc}$), and nuclear export rate ($r_{ex}$). We applied the Latin Hypercubic Sampling technique to reduce the sample size as required by the full sampling when multiple parameters were involved[28]. A parameter set that can consistently generate the WUS gradient in a similar range, quantified from the experimental data, was chosen as the baseline. The sensitivity analysis was performed by either decreasing the parameter values to 25% of the baseline or by increasing to 175% of the baseline. Each parameter range was divided into 25 uniform subintervals and 25 samples were chosen such that exactly one sample was drawn from each subinterval for each parameter. The output of the sensitivity analysis is the WUS nuclear concentration in L1 ($w_{n1}$), L2 ($w_{n2}$), and L3 ($w_{n3}$) layers, respectively. The partial correlation coefficient method was then applied to measure the pairwise correlation between each input and output, after removing the correlation due to other inputs. More specifically, to evaluate the correlation between $x_i$ and $y_j$, first, the linear regression analysis was performed between $x_i$ or $y_j$ and the rest of the inputs:

$$\hat{x}_i = a_0^i + \sum_{k=1, k\neq i}^{5} a_k^i x_k, \quad \hat{y}_j = b_0^j + \sum_{k=1, k\neq i}^{5} b_k^j x_k \quad (18)$$

Then the correlation between $x_i - \hat{x}_i$ and $y_j - \hat{y}_j$ was calculated to evaluate how sensitive the output $y_j$ is to the input $x_i$. The results of pairwise correlation between each input and output were shown in Supplementary Fig. 12a and the sensitivity of WUS nuclear level in each layer to the five parameters were shown in Supplementary Fig. 12b.

The results largely suggested that the nuclear WUS levels in L1, L2, and L3 were sensitive to all five parameters chosen as the inputs except to the WUS diffusion rate in the L1 layer at a significance level ≤ 0.01 (Supplementary Fig. 12a), showing the importance of the individual processes in regulating WUS nuclear levels. In detail, WUS nuclear levels in all three layers were positively correlated with mRNA synthesis rate, and negatively correlated with all other parameters, which is

biologically reasonable. A higher WUS synthesis can provide larger amounts of WUS proteins to migrate from deeper layers to outer layers. A higher diffusion rate across cells leads to a lower nuclear WUS level. A higher degradation rate (in nuclei or cytoplasm) leads to a lower level of protein in the nuclei. A higher nuclear export rate leads to a lower nuclear level. The results also suggested a decreasing trend in the sensitivity of almost all parameters from the L3 to the L1, except the nuclear degradation, indicating that the WUS nuclear level in the L3 is more sensitive to most of the biological processes studied. This could be because the full complement of all processes considered operates in the L3 cell layer. The nuclear WUS concentration in the L1 and to some extent in the L2 layer was found to be less sensitive to the diffusion rate. This could be due to, within the parameter range explored, very few proteins move into the outer layers irrespective of the diffusion rate. Overall, the sensitivity analysis with perturbation range ±75% on each parameter also confirmed that without the CLV3 and CK regulation, WUS protein gradient is significantly sensitive to WUS mRNA synthesis rate, WUS protein diffusion rate, nuclear and cytoplasmic degradation rate, and nuclear export rate, as shown in SI (Supplementary Fig. 12). Therefore, for the robust maintenance of the WUS gradient, extrinsic signals must act on one or several of the subcellular or cellular processes implicated in the WUS protein regulation. Hence, the CLV3-mediated post-translational regulation along with the transcriptional regulation of WUS together investigated in this work forms a potential mechanism for achieving robustness.

**Additional parameter sets to verify robustness mechanism**. The additional parameter sets out of the 220 samples from the Sobol Sampling method were selected based on the following criteria obtained from experimental quantifications:

1. The ratio of WUS nuclear protein between L3 and L1 was about 4 in *clv3* mutant and 3 in wild type (Supplementary Fig. 8h, i),
2. The N-C ratio in each layer was about 1 in both wild type and *clv3* mutant (Supplementary Fig. 8j),
3. Nuclear WUS protein accumulates in all cell layers including the outer L1 and L2 layers.
   The parameter sets that satisfied the ratio of WUS nuclear protein between L3 and L1 to be between 3 and 6, and N-C. ratios in each layer to be between 0.5 and 2, as well as the nuclear WUS to be greater than the threshold level (chosen to be 1 in this study) in L1 and L2 layers were selected. It was found that 4 out of 220 satisfied all these criteria (Supplementary Fig. 22). The other four satisfied the first two criteria about the ratios, however, they failed to accumulate WUS protein in the nuclei of the L1 and the L2 layers above the chosen threshold even without CLV3-mediated repression of *WUS* transcription (Supplementary Fig. 22). With the inclusion of CLV3-mediated repression of *WUS* transcription, the WUS protein levels reduced further rendering the system highly sensitive to obtain the balance between WUS and CLV3 in wild type. Two-dimensional plots of all parameter sets under the logarithmic scale (Supplementary Fig. 22), revealed that the selected parameter sets exhibited the following properties:

A. WUS cytoplasmic degradation rate increases linearly with the increase in diffusion rate, indicating $\sqrt{D_w/d_{wc}}$, which is the decay length of the WUS gradient, satisfies a constant. Since one of the selecting criteria restricts the fold change between L3 and L1, the spatial range of the WUS gradient becomes fixed within those selected parameter sets. Therefore, the correlation observed between the diffusion rate and the cytoplasmic degradation rate is consistent with the fact that the ratio between them $D_w/d_{wc}$ determines the spatial range of the WUS gradient.
B. The WUS nuclear export rate decreases linearly with the increase in the nuclear degradation rate. Such a negative correlation is required to regulate the N-C ratio (one of the selecting criteria) within a specific range and maintain the WUS gradient.
C. The WUS cytoplasmic degradation rate increases linearly with the increase in WUS production rate, which balances the total amount of WUS protein in all cell layers to maintain the WUS gradient. In contrast, the WUS nuclear degradation rate is not correlated with the production rate, which is expected because the nuclear WUS level is regulated by multiple processes including the nuclear import, export, and retention observed in experiments.

**Perturbation studies using additional parameter sets**. The four parameter sets obtained from global sensitivity analysis that satisfied all three criteria were further considered for calibration and perturbation studies. The analysis revealed that when *CLV3* is transcribed in the biologically relevant repression window of WUS concentration (Supplementary Fig. 13), the post-translational regulation along with the transcriptional regulation was able to maintain the normal *CLV3* expression domain in L1 and L2 layers over a large range of *WUS* mRNA production rates (Supplementary Figs. 15–18). Our simulations also revealed that the *CLV3* expression domain could be maintained with transcriptional regulation only when *CLV3* is transcribed within the activation window of WUS concentration (Supplementary Figs. 15–18). Theoretically, within the activation window, higher WUS

levels activate CLV3 which in turn represses *WUS* transcription, whereas lower WUS levels lead to loss of CLV3-mediated repression which restores *WUS* expression. However, such a mechanism is not biologically relevant because the wild-type *CLV3* expression is regulated within the repression window of WUS concentration.

**Reporting Summary**. Further information on research design is available in the Nature Research Reporting Summary linked to this article.

## Data availability
All data underlying this study are available from the corresponding author upon request because the full 3D confocal Airyscan images are large in size. Source data are provided with this paper.

## Code availability
An open-source MATLAB implementation of the computational model is available at GitHub (https://github.com/weitaoc/SAM/tree/main/CLV3_post_translational_regulation) or in Zenodo (https://doi.org/10.5281/zenodo.5527146)[29]. Source data are provided with this paper.

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

## Acknowledgements

This work was supported by the National Science Foundation Grants IOS-1456725 and 2055690, RSAP-AES mission funding to G.V.R., the National Science Foundation Grant DMS-1762063 through the joint NSF DMS/NIH NIGMS Initiative to M.A., G.V.R., and W.C., and the National Science Foundation Grant DMS-1853701 to W.C. We thank Dr. Stephen Snipes for quantification of the cytoplasmic levels of the EAR-like domain mutants of WUS.

## Author contributions

A.P. performed experiments, analyzed the data, and prepared the original draft. M.A. and K.R. contributed to the analysis and edited the manuscript. W.C. designed and performed computational analysis. G.V.R. conceived, designed, supervised this study, and wrote the manuscript. All the authors read and agreed to the manuscript.

## Competing interests

The authors declare no competing interests.
