## [Peer Review File · Nature Communications]

CLAVATA3 mediated simultaneous control of transcriptional and post-translational processes provides robustness to the WUSCHEL gradientREVIEWER COMMENTS

Reviewer #1 (Remarks to the Author):

The work by Plong et al. is a very interesting example of a good systems biology work in which experimental and computational work are likewise important and the modelling does not only play the role of window-dressing. The topic of the regulation of the stem cell niche and also the approach are not only relevant for specialised plant biologists but are also interesting for developmental biologists in general. Therefore I consider this manuscript in general well suited for NCOMM. Having said this, I have a few but important comments:

1. Almost half of the manuscript is about the analysis of the experimental findings using the mathematical model. The model plays therefore a central role. I would like to remind the authors that mathematical equations are a concise description of the model and NOT the diagram. The authors do a very good job to explain the model and the underlying assumptions in Fig. S6. I therefore strongly suggest to move this figure into the main text, maybe combined with the diagram in Fig. 2

2. My main concern with this manuscript is how the calibration was done, how the parameters were chosen and subsequently the sensitivity analysis was done

I do not believe that the data was by any means sufficient to uniquely determine the about 30 parameters of the model. If I am wrong here, I wish to see a proof. What I assume is that the parameters cannot be identified by the data, a very common problem in Systems Biology. But instead of exploring the parameter space in which the model behaves according to the data (the admissible parameter space), the authors decided to stick to a local analysis, without given a deeper justification for the parameters they have chosen. The authors base many important statements on the parameter set they have chosen, therefore it is very important to properly justify the approach/analysis. It can well be that all the statements hold for all/most of the admissible parameter space, but this needs to be shown. The Monte Carlo sampling strategy needs not only be applied to the sensitivity analysis, but also to the model behaviour (calibration). I understand this adds a very challenging dimension to the problem, but without, I fear, the modelling approach does not meet the standards required for a journal like NCOMM. I suggest not to use Latin Hypercube Sampling but Quasi Monte Carlo with low-discrepancy sequences like Sobol or Niederreiter. This can be parallelised very easily so the computational costs are affordable.

As I wrote in the beginning, I think this work is a valuable contribution. Unfortunately, it has some methodological deficiencies. I therefore do not go into any further details and I recommend that the authors re-work their manuscript. Without I cannot - unfortunately - recommend this work for publication in NCOMM.

Reviewer #2 (Remarks to the Author):

The manuscript by Alexander Plong¹ et al. describes the Confocal Laser Scanning Microscope analyses of WUS behavior at the shoot apical meristem of Arabidopsis. The authors analyzed WUS translocation into the nucleus and WUS degradation in WT and *clv3* mutant upon application of CLV3 peptide and Leptomycin-B, that inhibits nuclear export of protein into the cytoplasm.

Based on their analyses and previous published insights they developed mathematical models suggesting that CLV3 simultaneously controls WUS synthesis and sub-cellular partitioning.

In general, the question of what is the mechanism underlying the differential effect of a TF on cells, depend on their location, is fascinating and supper interesting for the broad community.

Regrettably the conclusion drawn in this manuscript are not well supported and therefore it is difficult to accept them.

The whole paper is based on analysis of GFP signals from the shoot apical meristem by confocal microscope. Unfortunately, few of the images presented seems not to be from median longitudinal sections and for some the quality is poor. The design of the experiments, lacking controls for confocal analysis (see below) and lacking molecular evidences, make it difficult to evaluate the results.

To summarize- to bring it to publication the authors must include controls, generate high quality image and add molecular evidences.

The following comments may help the authors improve their manuscript:

A. To determine the effect of CLV3-signaling on the WUS protein accumulation

the researcher followed the WUS protein reporter (*pWUS::eGFP-WUS*) in wild-type and *clv3-2* mutant backgrounds in response to exogenous application of the bioactive CLV3 peptide. The result of this experiment is presented in Fig 1.

The setting of the experiment is problematic and the strong conclusions are not well supported:

1. It seems that not all the sections of the meristem are median longitudinal sections and therefore any conclusion on WUS behavior is not accurate (for example Fig 1 K,L,M,O).
2. The quality of the confocal analysis is low. Since all of this work is based on the confocal analysis, it is better to analyze cryosections (Goldshmidt A et al, Plant Cell 2008).

3. There is no control! In order to conclude that WUS is degraded, the author must include controls. One control for example can be 35S::RFP-GR simultaneously with the 35S::GFP-WUS-GR. Ratio imaging analysis can be much more reliable to determine that WUS is degraded. (If you can see that the ratio is changed upon treatment or in the *clv3* mutant).

4. This study missing molecular evidence for degradation. If the authors do not include such evidence the confocal analysis must be more convincing.

B. " To experimentally test the requirement of active sensing of the WUS protein levels by CLV3 in regulating the WUS protein gradient, we exogenously supplied MCLV3 to the wild-type and the *clv3-2* plants carrying pWUS::eGFP-WUS in time-course experiments (Fig. 4A-H, S11A" " A similar treatment of the *clv3-2* mutants revealed a significant increase in WUS protein level in the L1 and a decrease in the deeper cell layers along with a severe reduction in WUS transcript level"

I am sure that the authors see the increase in WUS, but for the readers it is not sufficient. As said above: we don't see well the meristem and it is difficult to determine whether it is median section?; The florescent intensity seems to be different from one section to another; no controls are presented; and most important no molecular evidences – for example: RT-PCR to show the decrease in the WUS transcript, western blot of isolated nuclei as compared with total tissue etc.

To draw strong conclusion and construct a model you need more evidences or at least high quality images analysis

Minor remark:

1. You include in Fig 1 the treatment of sCLV3 with no explanation—what is this treatment? You must state in the legend that it is a control for the MCLV3

2. "The observed reduction of WUS protein levels in the L3 layers of wild-type and *clv3-2* SAMs is likely due to reduced WUS protein synthesis caused

by the repression of WUS transcription as shown in earlier studies^{15,16} (Fig. S2)".

"Fig. S2. WUS expression upon exogenous CLV3 peptide treatment"

- You can't show single time point (24h) and state that there is a reduction. To demonstrate a reduction, you must present time zero and the 24h

3. In several analyses presented here the authors used the 35S promoter to direct constitutive expression throughout the meristem. But according to their result (For example Fig 1K and N), the expression differs between the cell layer such that L1 and L2 exhibit very low signal.
4. Not in all figure the authors specified on what plant the analysis was performed on. For example, Fig S7— On which construct the analysis of GFP signal was done?
5. There is no BAR on Fig 4K

Reviewer #3 (Remarks to the Author):

The transcription factor WUS and a signaling pathway activated by CLV3 form a key regulatory loop in the shoot apical meristem (SAM). The ability of the SAM's central zone to maintain a consistent size depends on negative feedback: WUS promotes CLV3 expression while CLV3 inhibits WUS expression.

One of the more puzzling questions is: why does WUS promote CLV3 expression only in the two top layers of the meristem but not in the deeper layers where it is expressed? Two models currently exist. One group has proposed that WUS cannot activate CLV3 expression in the presence of the HAM transcription factor. The group of Dr. Reddy proposed that the function of WUS depends on its concentration: at higher concentrations WUS inhibits CLV3 expression while at lower concentrations it promotes it. But there is a problem with the second model – it creates an unstable system.

In an attempt to resolve this issue, in the current manuscript Plong and colleagues propose a novel function for the CLV3 signaling pathway: regulation of nuclear-cytoplasmic partitioning of WUS. Their conclusions are based on confocal analysis of WUS protein expression and on a computational model. However, there is a very limited amount of experimental data that on multiple occasions is missing statistical analysis, or statistical analysis suggests that observed differences are not significant (e.g. Fig. 1). The presented experimental data is insufficient to establish whether CLV3 regulates WUS post-transcriptionally or not.

While a model that integrates WUS synthesis with its sub-cellular distribution, diffusion, and degradation would be very valuable, the submitted model is based on weak experimental data and on questionable assumptions.

Specific comments related to the experimental data:

1. In the introduction the authors write: "WUS protein in the L1-layer fails to accumulate at a higher level despite a higher synthesis of the protein in the underlying cells (ref 9,12,13), suggesting that CLV3-signaling could play an additional role in the maintenance of the WUS protein in the CZ". WUS protein accumulation in *clv3* has been analyzed only in ref 9 and no quantitative comparison of WUS accumulation between different layers and between wt and *clv3* was done in that paper. Moreover, even if that assumption is true, I do not see how this suggests that "CLV3-signaling could play an additional role in the maintenance of the WUS protein". Why CLV3 and not some other pathway?

2. The following statements are not supported by data: "Seedlings treated with 1 μ M of the MCLV3, revealed only a slight reduction in WUS accumulation in the basal L3 layers" and "In contrast, the *clv3-2* mutant SAMs treated with MCLV3 revealed a more drastic reduction in WUS levels in the L3 and L2 layers, but an increase in the L1 layer and this trend was consistently observed over 6, 12, and 24 hrs of MCLV3 treatment." There is no statistically significant change after treatment with MCLV3 based on quantitative analysis of data as presented in Fig 1I and J. In Fig S1B, there is no statistically significant difference in L1 at 6h and 12 h and only a slight increase at 24 hours. Considering the size of error bars, I am surprised that this increase is statistically significant. It is not possible to make reliable conclusions based on Fig. 1A, C, D and F as there is always some variation of signal between individual SAMs. A stronger inhibitory effect of mCLV3 on expression of WUS in the internal layers is not surprising as that is where WUS is synthesized. Stronger evidence is necessary to prove that treatment with CLV3 leads to increased accumulation of WUS in the L1 layer.

3. The fluorescent signal in Fig1.K-T images should be quantified. In addition, the authors should compare eGFP-WUS-GR behavior to eGFP-GR. Is the potentially increased stability of eGFP-WUS-GP in response to CLV3 due to WUS or to GFP?

4. Fig.1 Treatment with DEX sends WUS to nucleus which leads to degradation of WUS. Treatment with LEP-B keeps WUS in the nucleus, but this does not alter stability of WUS. The authors do not explain why these two treatments, both increasing WUS concentration in nucleus, have different effects on WUS stability. Is it possible that DEX is less stable and after 24 h is not efficient in sending WUS to the nucleus? Testing control plants expressing eGFP-GR can help to answer this question.

5. Fig. 1 There are many additional explanations for the data besides CLV3 offsetting nuclear export of WUS at the posttranscriptional level. For example, the treatment with MCLV3 should increase expression of endogenous WUS. Formation of WUS - eGFP-WUS-GR dimers might stabilize eGFP-WUS-GR interaction with DNA which might lead to its retention in the nucleus. The authors should add cycloheximide to investigate whether observed changes are dependent on translation.

6. Page 6. "These observations show that the regulation of nuclear export is essential for maintaining the nuclear levels of WUS and suggest that CLV3-signaling may offset nuclear export." The presented data show that WUS can move in and out of the nucleus. But the data do not demonstrate that the nuclear export is regulated, much less demonstrating that this regulation is due to CLV3.

7. Page 7 "To distinguish between these possibilities, we challenged the cytoplasmically localized WUS (p35S::eGFP-WUS-GR) with exogenous application of MCLV3 (Fig. S5). Our results show that MCLV3

treatment did not change the cytoplasmic WUS protein levels in both wild type and *clv3-2*". There is no quantification of the data. Moreover, based on the submitted images, in the basal L3 tissues eGFP-WUS-GR seems to be moving to the nucleus in response to MCLV3.

8. Fig S2. qRT-PCR should be performed instead of semi-quantitative PCR.

9. Fig S3P. Can the authors comment on why mutation of nuclear export signal leads to WUS accumulation in the cytosol? Should it not be otherwise?

Specific comments related to the computational model:

Several parts of the model involve fit-for-purpose and nonstandard forms of equations that do not seem to derive from biochemical principles. I don't object to phenomenological models, but clear descriptions and sound justifications must be included.

1. It looks like the authors assumed that the lateral restriction of CLV3 expression is due to the inhibition of lateral WUS diffusion by CLV3. The model basically has a 'highway' in the center of the SAM with lateral barriers (soundwall?). How can CLV3 only inhibit the lateral diffusion? Does it specifically recognize the plasmodesmata in the lateral direction? Also, this crucial assumption about lateral inhibition is not included in the listed equations.

2. Fig S6. The equation for C_P has a nonstandard form of nonlinear function that lacks mechanistic basis. Why can't the biphasic regulation of CLV3 by WUS be modeled with multiplication of Hill functions like other gene regulations?

3. Most of the Hill coefficients are extremely high (10). What are the possible reasons for this high nonlinearity from a mechanistic point of view? For gene regulations this may be explained by high numbers of binding sites, but for processes like diffusion and nuclear transport, it's difficult to conceive the mechanisms.

4. The parameter values for restricting expressing domains of WUS and CLV3 (e.g. radius) are not listed. The domain restriction of CK receptor is not clearly defined in the equations.

5. The threshold parameters in Hill functions are known as EC_{50} , not EZ_{50} .

REVIEWER COMMENTS

Reviewer #1 (Remarks to the Author):

The work by Plong et al. is a very interesting example of a good systems biology work in which experimental and computational work are likewise important and the modelling does not only play the role of window-dressing. The topic of the regulation of the stem cell niche and also the approach are not only relevant for specialised plant biologists but are also interesting for developmental biologists in general. Therefore I consider this manuscript in general well suited for NCOMM. Having said this, I have a few but important comments:

1. Almost half of the manuscript is about the analysis of the experimental findings using the mathematical model. The model plays therefore a central role. I would like to remind the authors that **mathematical equations are a concise description of the model and NOT the diagram**. The authors do a very good job to explain the model and the underlying assumptions in Fig. S6. I therefore strongly suggest to move this figure into the main text, maybe combined with the diagram in Fig. 2

Response: We thank the reviewer for appreciation of the importance of the modeling effort in our paper. Thank you for the suggestion. We have now moved the model diagram and description of the underlying assumptions from Fig. S6 into the main text Fig. 5 (see Page 38, Line 763).

2. My main concern with this manuscript is how the calibration was done, how the parameters were chosen and subsequently the sensitivity analysis was done.

I do not believe that the data was by any means sufficient to uniquely determine the about 30 parameters of the model. If I am wrong here, I wish to see a proof. What I assume is that the parameters cannot be identified by the data, a very common problem in Systems Biology. But instead of exploring the parameter space in which the model behaves according to the data (the admissible parameter space), the authors decided to stick to a local analysis, without given a deeper justification for the parameters they have chosen. The authors base many important statements on the parameter set they have chosen, therefore it is **very important to properly justify the approach/analysis**. It can well be that all the statements hold for all/most of the admissible parameter space, but this needs to be shown. The Monte Carlos sampling strategy needs not only be applied to the sensitivity analysis, but also to the model behaviour (calibration). I understand this adds a very challenging dimension to the problem, but without, I fear, the modelling approach does not meet the standards required for a journal like NCOMM. I suggest not to use Latin Hypercube Sampling but **Quasi Monte Carlo** with low-discrepancy sequences like **Sobol or Niederreiter**. This can be parallelised very easily so the computational costs are affordable.

Response: We thank the reviewer for this comment. As suggested, we have performed calibration of the full model by performing a Quasi Monte Carlo analysis

for the basic PDE model and analysis of the steady state behavior of the corresponding ODE model. We added a new subsection, titled ‘Calibration of the Computational Model’, in the Methods Section in the revised manuscript (see Page 23, Line 490). The new results are provided in the Supplementary Figure S12 (see Page 59, Line 938). The local sensitivity analysis results obtained earlier were moved to Figure S13 (see Page 61, Line 956). Below are the answers to specific comments based on the new analysis.

My main concern with this manuscript is how the calibration was done, how the parameters were chosen and subsequently the sensitivity analysis was done. I do not believe that the data was by any means sufficient to uniquely determine the about 30 parameters of the model.

In the original submission, we did not provide a detailed description of the calibration of the computational model. Only local sensitivity analysis was performed within a certain range around a selected parameter set which could generate consistent results with the experimental data. We now followed the suggestion by the reviewer and implemented the Quasi Monte Carlo simulations with Sobol sampling over a much larger parameter space for the basic PDE model, followed by a steady state analysis of the corresponding ODE model, to determine the parameters associated with signaling regulations to complete the calibration.

The following text was added in the revised manuscript on Page 23, Line 490.

Calibration of the Basic PDE Model. We consider the following basic PDE model to capture the dynamics of WUS mRNA and protein in different subcellular compartments without any extrinsic signaling regulations:

$$\begin{aligned}\frac{\partial[W]}{\partial t} &= W_p(x, y, z) - d_w[W] \\ \frac{\partial[W_c]}{\partial t} &= D_w \Delta_w [W_c] + r_c[W] - d_{wc}[W_c] + r_{ex}[W_n] - r_{im}[W_c] \\ \frac{\partial[W_n]}{\partial t} &= -d_{wn} \left(d_{\min} + \frac{d_{\max} - d_{\min}}{\left(1 + ([W_n]/k_{ww})^{n_z}\right)} \right) [W_n] - r_{ex}[W_n] + r_{im}[W_c]\end{aligned}$$

$$W_p(x, y, z) = A_1 \cdot I_{\{x^2+y^2 \leq r_w^2, z \leq L_w\}}(x, y, z)$$

This basic PDE model is helpful to estimate the effective diffusion rate, nuclear export rate, import rate and degradation rates with the feedback regulations in the full model. Here are the ranges for the critical parameters used in the Sobol sampling:

WUS mRNA synthesis rate: [0.1, 100]

Diffusion rate of WUS protein: [0.01, 10]

Degradation rate of WUS nuclear protein: [0.01, 10]

Degradation rate of WUS cytoplasmic protein: [0.01, 10]

Nuclear export rate of WUS protein: [0.01, 10]

Since the Sobol sampling was performed over a large range across different magnitudes, the random samples were obtained after taking the logarithmic function at base 10 for each parameter and then transformed back to the exact ranges. In the revision, we ran 200 simulations for two different random samples as well as the mixed parameter sets between them as required when calculating the sensitivity indexes following (Andrea Saltelli 2002, Computer Physics Communications 145 (2) pp. 280-297). More specifically, two random samples M1 and M2 were generated with the same sample size following Sobol sequencing independently from the same ranges by using the MATLAB function *sobolset*:

$$M_1 = \begin{pmatrix} \vdots \\ x_1 & x_2 & x_3 & x_4 & x_5 \\ \vdots \end{pmatrix}, M_2 = \begin{pmatrix} \vdots \\ x'_1 & x'_2 & x'_3 & x'_4 & x'_5 \\ \vdots \end{pmatrix}$$

The total effect indices were defined as

$$\hat{S}_j^i = 1 - \frac{\hat{U}_{-j}^i - \hat{E}^2(y_i)}{\hat{V}(y_i)}$$

where y_i was the i -th output, n was the sample size, j denoted the j -th input, and

$$\hat{E}^2(y_i) = \frac{1}{n} \sum_{r=1}^n y_i(x_{r1}, x_{r2}, x_{r3}, x_{r4}, x_{r5}) y_i(x'_{r1}, x'_{r2}, x'_{r3}, x'_{r4}, x'_{r5})$$

$$\hat{U}_{-j}^i = \frac{1}{n-1} \sum_{r=1}^n y_i(x_{r1}, x_{r2}, x_{r3}, x_{r4}, x_{r5}) y_i(x_{r1}, \dots, x'_{rj}, \dots, x_{r5})$$

$$\hat{V}(y_i) = \frac{1}{n} \sum_{r=1}^n y_i^2(x_{r1}, x_{r2}, x_{r3}, x_{r4}, x_{r5}) - \left[\frac{1}{n} \sum_{r=1}^n y_i(x_{r1}, x_{r2}, x_{r3}, x_{r4}, x_{r5}) \right]^2$$

Here the index S_j^i represented the total effect of the j -th input on the i -th output. In this analysis, we computed the total effect indices of all five inputs on the WUS nuclear levels at L1, L2 and L3, respectively. The global sensitivity analysis showed that the WUS nuclear level in each of three layers had positive correlation with the WUS mRNA synthesis rate and negative correlation with all the other parameters.

This was shown to be biologically relevant in Figure S12A in the revised SI (Page 59, Line 938).

By using the Quasi Monte Carlo sensitivity analysis to investigate properties of the basic PDE model which involved much less parameters, we were able to select proper parameter sets that could generate results more consistent with the experimental data. In particular, we concluded that the appropriate criteria to select the parameter values were as follows: WUS nuclear protein level in L3 was about three fold of that in L1 and the ratio between WUS protein concentrations in nucleus and cytoplasm was close to 1 across different layers.

Steady State of the Corresponding ODE System. Next, we calibrated the parameters associated with WUS-CLV3 feedback regulations by considering the following ordinary differential equation system and its steady state, which is close to the steady state obtained in the original PDE model by ignoring the spatial distribution:

$$\begin{aligned} \frac{d[W]}{dt} &= \frac{W_p}{1 + ([CLV3]/k_{CW}^1)^{n_1}} - d_w[W] \\ \frac{d[W_c]}{dt} &= r_c[W] - d_{wc}[W_c] + \frac{r_{ex}}{\left(1 + ([CLV3]/k_{CW}^2)^{n_2}\right)} [W_n] - r_{im}[W_c] \\ \frac{d[W_n]}{dt} &= -d_{wn} \left(d_{min} + \frac{d_{max} - d_{min}}{\left(1 + ([W_n]/k_{ww})^{n_3}\right)} \right) [W_n] - \frac{r_{ex}}{\left(1 + ([CLV3]/k_{CW}^2)^{n_2}\right)} [W_n] + r_{im}[W_c] \\ \frac{d[CLV3]}{dt} &= \frac{C_p}{\left(1 + ([W_n]/k_{WC}^1)^{n_4}\right) \left(1 + ([W_n]/k_{WC}^2)^{-n_5}\right)} - d_c[CLV3] \end{aligned}$$

The system could be simplified into one nonlinear equation and solved using some numerical method. More specifically, the steady state solutions were obtained by solving the following equations:

$$\begin{aligned} [CLV3]_{ss} &= \frac{C_p/d_c}{\left(1 + ([W_n]_{ss}/k_{WC}^1)^{n_4}\right) \left(1 + ([W_n]_{ss}/k_{WC}^2)^{-n_5}\right)} \\ [W]_{ss} &= \frac{W_p/d_w}{1 + ([CLV3]_{ss}/k_{CW}^1)^{n_1}} \\ [W_c]_{ss} &= \frac{d_{wn}}{r_{ex}} \left(d_{min} + \frac{d_{max} - d_{min}}{\left(1 + ([W_n]_{ss}/k_{ww})^{n_3}\right)} \right) [W_n]_{ss} + \frac{r_{ex}/r_{im}}{\left(1 + ([CLV3]_{ss}/k_{CW}^2)^{n_2}\right)} [W_n]_{ss} \\ r_c[W]_{ss} - d_{wc}[W_c]_{ss} + \frac{r_{ex}}{\left(1 + ([CLV3]_{ss}/k_{CW}^2)^{n_2}\right)} [W_n]_{ss} - r_{im}[W_c]_{ss} &= 0 \end{aligned}$$

and the last nonlinear equation was solved numerically in MATLAB by the function *fzero*. Notice that there existed a unique stable steady state for this ODE model

such that W_n and $CLV3$ coexisted. This allowed us to calibrate the parameters, especially the EC50 coefficients in Hill functions which modeled specific regulations, by doing a local perturbation of each single parameter on the stable steady state (Figure S12B, Page 59, Line 938). We also investigated the model by removing the post-translational regulation of $CLV3$ on WUS protein. The number of steady states remained the same. However, the steady state which had W_n and $CLV3$ coexisting and was stable in the system with dual regulations, became unstable under the transcriptional regulation only. This suggested that the post-translational regulation helped the WUS - $CLV3$ system reach a stable equilibrium with robustness. The results were provided in Figure S12C-D (Page 59, Line 938) in the revised Supplementary.

Again the parameters were chosen such that the same criteria were satisfied and WUS protein and $CLV3$ could coexist in the steady state of the system. By integrating the results of Quasi Monte Carlo sensitivity analysis and the steady state analysis of the corresponding ODE system, we were able to calibrate most of the parameters. This resulted in an appropriate parameter set for the full model that could generate simulation results close to the experimentally observed protein levels in both nucleus and cytoplasm in different layers.

It can well be that all the statements hold for all/most of the admissible parameter space, but this needs to be shown.

We thank the reviewer for this important comment. It is very challenging to verify whether the hypothesized mechanism could work for the entire admissible parameter space for the full very complex biological system. However, we are able to show that within certain ranges of biologically relevant parameters, the system behavior is robust under the proposed mechanism. Mechanism and parameter ranges predicted by model simulations can be verified in future experiments on measuring some of these parameters including export rate and diffusion rates. However, experimental determination of those rates is challenging and requires further studies.

As I wrote in the beginning, I think this work is a valuable contribution. Unfortunately, it has some methodological deficiencies. I therefore do not go into any further details and I recommend that the authors re-work their manuscript. Without I cannot - unfortunately - recommend this work for publication in NCOMM.

Reviewer #2 (Remarks to the Author):

The manuscript by Alexander Plong1 et al. describes the Confocal Laser Scanning Microscope analyses of WUS behavior at the shoot apical meristem of Arabidopsis. The authors analyzed WUS translocation into the nucleus and WUS degradation in WT and *clv3* mutant upon application of CLV3 peptide and Leptomycin-B, that inhibits nuclear export of protein into the cytoplasm. Based on their analyses and previous published insights they developed mathematical models suggesting that CLV3 simultaneously controls WUS synthesis and sub-cellular partitioning. In general, the question of what is the mechanism underlying the differential effect of a TF on cells, depend on their location, is fascinating and super interesting for the broad community.

Response: We thank the reviewer for the positive comments. We have now revised the manuscript based on the suggestions provided which has improved the manuscript quality and provides stronger support for the earlier proposed model.

Regrettably the **conclusions drawn in this manuscript are not well supported** and therefore it is difficult to accept them.

The whole paper is based on analysis of GFP signals from the shoot apical meristem by confocal microscope. Unfortunately, few of the images presented seems not to be from median longitudinal sections and for some the quality is poor. The design of the experiments, lacking controls for confocal analysis (see below) and lacking molecular evidences, make it difficult to evaluate the results.

To summarize- to bring it to publication the authors must include controls, generate high quality image and add molecular evidences.

The following comments may help the authors improve their manuscript:

A. To determine the effect of CLV3-signaling on the WUS protein accumulation the researcher followed the WUS protein reporter (pWUS::eGFP-WUS) in wild-type and *clv3-2* mutant backgrounds in response to exogenous application of the bioactive CLV3 peptide. The result of this experiment is presented in Fig 1.

The **setting of the experiment is problematic** and the strong **conclusions are not well supported**:

1. It seems that not all the sections of the meristem are median longitudinal sections and therefore any conclusion on WUS behavior is not accurate (for example Fig 1 K,L,M,O).

Response: To address this comment we have obtained new higher magnification images at improved resolution which has enhanced image quality. The sections shown in all figures are indeed the median longitudinal sections and we have now

provided serial sections from one image stack to illustrate the point (See Fig. S16; Page 64, Line 993).

2. The quality of the confocal analysis is low. Since all of this work is based on the confocal analysis, it is better to analyze cryosections (Goldshmidt A et al, Plant Cell 2008).

Response: We thank the reviewer for suggesting fixing the tissues followed by cryosectioning. However, upon fixing, Arabidopsis SAMs revealed an enormous amount of autofluorescence which completely masked the eGFP-WUS signal and prevented us from quantifying the signal (Fig. S10A; Page 56, Line 910). Therefore, we have managed to acquire better quality images without fixing which have been used for quantification of WUS protein signal and the new data has been provided (Figs. 1-4). The original Fig. 1 which showed the behaviors of both the *pWUS::eGFP-WUS* and *35S::eGFP-WUS-GR* has been split up into four figures; the new Fig. 1 (Page 34, Line 707) and Fig. 2 (Page 35, Line 723) show the effects of the short and long term MCLV3 treatments on *pWUS::eGFP-WUS* respectively, and the new Fig. 4 (Page 37, Line 752) shows the effect of LMB treatments. The Fig. 3 (Page 36, Line 737) now shows the effects of MCLV3 and LMB treatments on *35S::eGFP-WUS-GR*.

In addition, we have compared our previously estimated N-C ratios of *pWUS::eGFP-WUS* in wild-type with the new analysis that included double labeling with DAPI. Since DAPI staining requires tissue fixing which produces high levels of autofluorescence, we first acquired eGFP-WUS images and the same SAMs were fixed and stained with DAPI. The images were registered post-acquisition by using Photoshop to align the DAPI signal with WUS with the nuclei and cell boundary as landmarks to align the Z-stacks. The N-C ratios obtained by this new analysis closely matches with the quantification presented in earlier version of the manuscript which did not use DAPI counterlabeling (See Fig. S10D-G; Page 56, Line 910).

3. There is no control! In order to conclude that WUS is degraded, the author must include controls. One control for example can be *35S::RFP-GR* simultaneously with the *35S::GFP-WUS-GR*. Ratio imaging analysis can be much more reliable to determine that WUS is degraded. (If you can see that the ratio is changed upon treatment or in the *clv3* mutant).

Response: We thank the reviewer for this comment. We have shown in our earlier work that *35S::GFP-GR* fails to degrade as that of *35S::eGFP-WUS-GR* upon nuclear translocation showing that this behavior is specific to WUS and not GFP (Rodriguez et al. 2016, PNAS 113 (41) pp. E6307-E6315). We introduced the same construct into *clv3-2* and analyzed its behavior upon dex and MCLV3 treatments. Our analysis shows that both in the wild type and in *clv3-2* mutants, the *35S::eGFP-GR* is not degraded upon its dex-mediated translocation into the nucleus. Moreover, MCLV3 treatments did not change the GFP protein accumulation showing that the observed effects are specific to WUS (Fig. S1; Page 42, Line 817). Since our

quantification clearly reveals drastic differences between 35S::eGFP-WUS-GR (Fig. 3 and S2; Page 36-Line 737 and Page 44-Line 831) and 35S::eGFP-GR (Fig. S1; Page 42, Line 817), we did not attempt the ratiometric analysis which requires the development of RFP-GR and the RFP being a different fluorescent protein may not be directly comparable to the eGFP behavior.

4. This study missing molecular evidence for degradation. If the authors do not include such evidence the confocal analysis must be more convincing.

Response: We thank the reviewer for this comment. We agree that the identity of the degradation signal remains elusive. Therefore, as suggested by the reviewer, we have improved the image quality and the new images have been used for quantification which agree with our earlier analysis on the effects of MCLV3 and Leptomycin on *pWUS::eGFP-WUS* and 35S::eGFP-WUS-GR in wild type and in *clv3-2* mutants (Fig. 1-4; Pages 34-37).

B. " To experimentally test the requirement of active sensing of the WUS protein levels by CLV3 in regulating the WUS protein gradient, we exogenously supplied MCLV3 to the wild-type and the *clv3-2* plants carrying *pWUS::eGFP-WUS* in time-course experiments (Fig. 4A-H, S11A" " A similar treatment of the *clv3-2* mutants revealed a significant increase in WUS protein level in the L1 and a decrease in the deeper cell layers along with a severe reduction in WUS transcript level"

I am sure that the authors see the increase in WUS, but for the readers it is not sufficient. As said above: we don't see well the meristem and it is difficult to determine whether it is median section?; The florescent intensity seems to be different from one section to another; no controls are presented; and most important no molecular evidences – for example: RT-PCR to show the decrease in the WUS transcript, western blot of isolated nuclei as compared with total tissue etc.

To draw strong conclusion and construct a model you need more evidences or at least high quality images analysis

Response: We thank the reviewer for this comment. We have now provided the WUS protein quantification from improved images and the qRT-PCR data on WUS transcript levels for all time points, both in wild type and *clv3-2* mutants (Fig. 1-2, S3; Pages 34-35, 45).

With the *pWUS::eGFP-WUS*, it should be noted that it is a fight between lack of continued production of WUS due to a rapid loss of *WUS* transcription and nuclear accumulation of the WUS protein upon MCLV3 peptide treatment. Therefore, to uncouple or to mitigate the effect of MCLV3 on *WUS* transcription, we have measured the effect of MCLV3 on 35S::eGFP-WUS-GR (the best system that we can use as it combines transient translocation of WUS, the heterologous 35S promoter and also directly provides WUS to the nuclei of cells in all layers to mitigate the complexities associated with the protein mobility that is in turn linked to N-C partitioning). This analysis also revealed that MCLV3 or LMB treatment leads to a

stable nuclear accumulation of WUS supporting the results obtained from analysis using the *pWUS::eGFP-WUS*.

With regard to the use of immunoblot on isolated nuclei, the precise experiment has been carried out in our earlier work (see Fig. S2B in Yadav et al. 2011, *Genes and Development* 25 (19) pp. 2025-2030). The WUS protein was still detectable in the nuclear extracts of SAMs overexpressing ethanol inducible CLV3 in *clv3-2* background, even after 48hrs of exposure to ethanol. The immunoblot experiment and the imaging of *pWUS::eGFP-WUS* presented in this study revealed WUS protein continues to accumulate despite a rapid loss of *WUS* transcription as early as 30 min into MCLV3 application (Lee et al. 2019; *Plant Cell Rep.* (2019) Mar; 38 (3) pp. 311-319) and remains severely downregulated within the 3-72 hrs window tested here (Fig. 1I-K, 2I-J; Pages 34-35).

Moreover, the continued accumulation of the WUS protein in MCLV3 treated plants must be viewed in the context of a very short half-life of the WUS protein especially in outer cell layers in WT and even higher instability observed in *clv3-2* mutants. Despite rapid loss of *WUS* transcription and rapid WUS protein turn over, our quantification of *pWUS::eGFP-WUS* in the L1 layer of *clv3-2* mutants revealed a statistically significant increase till 48 hrs after MCLV3 treatment (Fig. 2; Page 35), showing the importance of CLV3 signaling in stabilizing the WUS protein (See point #1; reviewer#3 for arguments on WUS protein turnover and the importance of CLV3-mediated stabilization of WUS in SAM maintenance).

Minor remark:

1. You include in Fig 1 the treatment of sCLV3 with no explanation—what is this treatment? You must state in the legend that it is a control for the MCLV3

Response: Thank you for the comment. Our apologies. We have now added in our Methods (Page 17, Line 348) and Figure captions (Page 45, Line 838) that the scrambled peptide (sCLV3) is the control, which is the scrambled version of the bioactive MCLV3 peptide for treatments (Table S4; Page 70).

2. "The observed reduction of WUS protein levels in the L3 layers of wild-type and *clv3-2* SAMs is likely due to reduced WUS protein synthesis caused by the repression of WUS transcription as shown in earlier studies 15,16 (Fig. S2)".

"Fig. S2. WUS expression upon exogenous CLV3 peptide treatment"

• You can't show single time point (24h) and state that there is a reduction. To demonstrate a reduction, you must present time zero and the 24h.

Response: Thank you for this comment. We have updated our analysis to match the length of the time-course from our imaging experiments which revealed severe downregulation of *WUS* transcription in the 3-72hr window (Figs. 1-2, S3; Pages 34-35, 45).

3. In several analyses presented here the authors used the 35S promoter to direct constitutive expression throughout the meristem. But according to their result (For example Fig 1K and N), the expression differs between the cell layer such that L1 and L2 exhibit very low signal.

Response: Thank you for this comment. Note that we have preselected 35S::eGFP-WUS-GR for relatively uniformly expressing lines screening nearly 24 independent lines as described in Rodriguez et al. 2016, PNAS 113 (41) pp. E6307-E6315). Despite this effort, we have observed slightly lower GFP fluorescence in the L1 and the L2 compared to the L3. However, the drastically lower WUS protein accumulation within 24hrs of dex treatment in wild type and 6hrs in *clv3-2* mutants, and the significant recovery of WUS protein levels upon MCLV3 and LMB treatments show that small expression differences of the 35S promoter should not affect the results and the conclusions.

A higher GFP fluorescence in the L3/pith of *clv3-2* SAMs was observed both with 35S::eGFP-WUS-GR and 35S::eGFP-GR. Despite the higher levels, the dex-mediated nuclear translocation led to a dramatic destabilization of 35S::eGFP-WUS-GR, including the L3 layers, of *clv3-2* mutants showing that the nuclear translocation of eGFP-WUS-GR leads to a dramatic destabilization of the protein (Fig. S2; Page 44).

4. Not in all figure the authors specified on what plant the analysis was performed on. For example, Fig S7— On which construct the analysis of GFP signal was done?

Response: Thank you for this comment. We have now split up the figures, for example Fig. 1, Fig. 2, and Fig. 4 shows the behavior of *pWUS::eGFP-WUS* and Fig. 3 shows the behavior of 35S::eGFP-WUS-GR. We have also updated the legend to specify which construct was being analyzed in the improved Fig. S10.

5. There is no BAR on Fig 4K

Response: Thank you. We have included the scale bar in image 4K which is now the image S15E (Page 63).

Reviewer #3 (Remarks to the Author):

The transcription factor WUS and a signaling pathway activated by CLV3 form a key regulatory loop in the shoot apical meristem (SAM). The ability of the SAM's central zone to maintain a consistent size depends on negative feedback: WUS promotes CLV3 expression while CLV3 inhibits WUS expression.

One of the more puzzling questions is: why does WUS promote CLV3 expression only in the two top layers of the meristem but not in the deeper layers where it is expressed? Two models currently exist. One group has proposed that WUS cannot activate CLV3 expression in the presence of the HAM transcription factor. The group of Dr. Reddy proposed that the function of WUS depends on its concentration: at higher concentrations WUS inhibits CLV3 expression while at lower concentrations it promotes it. But there is a problem with the second model – it creates an unstable system.

In an attempt to resolve this issue, in the current manuscript Plong and colleagues propose a novel function for the CLV3 signaling pathway: regulation of nuclear-cytoplasmic partitioning of WUS. Their conclusions are based on confocal analysis of WUS protein expression and on a computational model. However, there is a very limited amount of experimental data that on multiple occasions is missing statistical analysis, or statistical analysis suggests that observed differences are not significant (e.g. Fig. 1). The presented experimental data is insufficient to establish whether CLV3 regulates WUS post-transcriptionally or not.

While a model that integrates WUS synthesis with its sub-cellular distribution, diffusion, and degradation would be very valuable, the submitted model is based on **weak experimental data and on questionable assumptions**.

Response: Thank the reviewer for comments. Now we have answered all questions raised with new experimentation and analysis which support the new model developed here. Rationalization with the HAM-WUS interaction model is possible only after the examination of WUS protein accumulation in *ham* mutants which is lacking at the moment. What if HAMS are required for maintaining the WUS protein stability? The instability/higher WUS protein turn over in *ham* mutants could result in activation of CLV3 in the RM.

Specific comments related to the experimental data:

1. In the introduction the authors write: “WUS protein in the L1-layer fails to accumulate at a higher level despite a higher synthesis of the protein in the underlying cells (ref 9,12,13), suggesting that CLV3-signaling could play an additional role in the maintenance of the WUS protein in the CZ”. WUS protein accumulation in *clv3* has been analyzed only in ref 9 and no quantitative comparison of WUS accumulation between different layers and between wt and *clv3* was done in that paper. Moreover, even if that assumption is true, I do not see how this

suggests that “CLV3-signaling could play an additional role in the maintenance of the WUS protein”. Why CLV3 and not some other pathway?

Response: Thank you for the comment. The quantification of WUS protein in different cell layers of the wild type and the *clv3-2* mutants were provided in the earlier version of the manuscript and this quantification has been improved now with new images (Fig. S10F-G; Page 56). The data shows that despite a higher WUS synthesis in the deeper cell layers of *clv3-2* mutants, the protein fails to build up in the L1 cell layer. A similar lack of WUS protein accumulation in the nuclei of the L1 layer cells in *clv3* mutants has been found in earlier study using the immunohistochemical analysis (Daum et al. 2014; PNAS 111 (4) pp. 14619-14624). This is the reason to hypothesize that CLV3 signaling could play additional roles in regulating the WUS protein levels and to proceed with MCLV3 peptide treatments. The new data of the finer time scale of MCLV3 peptide treatment analysis and improved images reveal a significantly higher WUS protein accumulation in the L1 layer of *clv3-2* mutants despite a drastic reduction in *WUS* transcription (see Fig. 11-K; Page 34). Since the system is responding to MCLV3 peptide treatment in maintaining WUS protein levels for several days after WUS transcription is turned off (see point#2), it shows that CLV3 signaling is required for maintaining nuclear levels of the WUS protein and SAM growth

Yes there could be other signals stabilizing WUS as our earlier work has shown that cytokinin in the rib meristem stabilizes WUS by acting on the nuclear retention signal (Snipes et al. 2018; PLOS Genetics 14 (4) pp. e1007351) which has been included in the model. Also note that our earlier work shows that cytokinin response visualized with pTCS::GFP remains localized to the L3 layers both in *clv3-2* (See Fig. 1E, Supplemental Fig. S3C in Snipes et al. 2018; PLOS Genetics 14 (4) pp. e1007351) and *wus-1* mutants (See Fig. 6G in Snipes et al. 2018; PLOS Genetics 14 (4) pp. e1007351), therefore it is regulated independently of these two regulators. This rules out the possibility that MCLV3 induced WUS protein stability could be due to the changes in cytokinin response.

2. The following statements are not supported by data: “Seedlings treated with 1 μ M of the MCLV3, revealed only a slight reduction in WUS accumulation in the basal L3 layers” and “In contrast, the *clv3-2* mutant SAMs treated with MCLV3 revealed a more drastic reduction in WUS levels in the L3 and L2 layers, but an increase in the L1 layer and this trend was consistently observed over 6, 12, and 24 hrs of MCLV3 treatment.” There is no **statistically significant change after treatment with MCLV3 based on quantitative analysis of data as presented in Fig 1I and J**. In Fig S1B, there is no statistically significant difference in L1 at 6h and 12 h and only a slight increase at 24 hours. Considering the size of error bars, I am surprised that this increase is statistically significant. It is not possible to make reliable conclusions based on Fig. 1A, C, D and F as there is always some variation of signal between individual SAMs. *A stronger inhibitory effect of mCLV3 on expression of WUS in the internal layers is not be surprising as that is where WUS is synthesized.* **Stronger evidence is necessary to prove that treatment with CLV3 leads to increased accumulation of WUS in the L1 layer.**

Response: Thank you for the comment. New quantification along with statistical analysis carried out on newly acquired and improved images has been provided. Quantified data that now also includes a new 3hr time point to capture the WUS protein dynamics before the WUS protein levels drop dramatically as a result of instantaneous shut down of *WUS* transcription upon MCLV3 treatments. We have organized data into short term (Fig. 1G-H; Page 34) and the long term treatments (Fig. 2K-L; Page 35). Note that an earlier study has shown that MCLV3 peptide treatment shuts down *WUS* transcription within 30 minutes of treatment (Lee et al. 2019; *Plant Cell Rep.* (2019) Mar; 38 (3) pp. 311-319). However, the WUS protein continues to accumulate in SAMs even after 72 hrs of MCLV3 peptide treatment which clearly shows the dramatic stabilizing effect of CLV3 on the WUS protein.

This stabilizing effect mediated by the MCLV3 must also be viewed in the context of extreme instability/high turnover of WUS protein in the outer cell layers of the SAM that could be inferred from these observations. a) Blocking WUS movement by callose induction for 8 hrs led to no WUS protein accumulation in outer cell layers (Daum et al. 2014; PNAS 111 (4) pp. 14619-14624). b) The nuclear translocation of WUS by using the 35S::eGFP-WUS-GR leads to its degradation in the CZ within 6hrs of dex application in wild type plants (See Fig. 4 in Rodriguez et al. 2016, PNAS 113 (41) pp. E6307-E6315) and even faster degradation within 3hrs in *clv3-2* mutants (Fig. 3; Page 36), suggesting that WUS protein is highly unstable and its half-life may be in the order of 3 hours or less. Considering such a short half-life, the maintenance of WUS protein for at least 72 hrs after MCLV3 peptide treatment (Fig. 2; Page 35), despite rapid loss of *WUS* transcription as early as 30 minutes into the treatment (Lee et al. 2019; *Plant Cell Rep.* (2019) Mar; 38 (3) pp. 311-319; Fig. 1), provides a strong evidence for post-transcriptional role for CLV3 signaling in stabilizing the WUS protein. This prolonged accumulation of the WUS protein also provides an explanation for the continued maintenance of the SAM growth for several days, even at extremely high levels of CLV3 achieved by using CLV3 promoter variants (Muller et al. 2006; *The Plant Cell* 18 (5) pp. 1188-1198) or by using inducible systems to activate CLV3 (Yadav et al. 2010; *Development* 137 (21) pp. 3581-3589).

To uncouple the effect of CLV3 signaling on *WUS* transcription from that of the WUS protein, we used the WUS expressed from the heterologous system (35S::eGFP-WUS-GR) which clearly shows that nuclear translocated WUS can accumulate stably in plants supplemented with MCLV3 showing the effects of CLV3 in stabilizing the WUS protein (See also point #4; Reviewer#2).

3. The fluorescent signal in Fig1.K-T images should be quantified. In addition, the authors should compare eGFP-WUS-GR behavior to eGFP-GR. Is the potentially increased stability of eGFP-WUS-GP in response to CLV3 due to WUS or to GFP?

Response: Thank you for the comment. The quantified data has been provided now. The behavior of 35S::eGFP-GR both in wild type (which was also provided in our earlier study; Rodriguez et al. 2016, PNAS 113 (41) pp. E6307-E6315) and in *clv3-2* mutants has been provided now (See point #3; reviewer #2; Figures 3 and

Supplementary Figure 1 on Page 36 and 42, respectively). The analysis shows that eGFP-GR accumulates stably after dex treatment and the levels are not influenced by MCLV3. Therefore, the destabilization of eGFP-WUS upon its nuclear translocation and stabilization upon MCLV3 treatment is specific to WUS and not GFP.

4. Fig.1 Treatment with DEX sends WUS to nucleus which leads to degradation of WUS. Treatment with LEP-B keeps WUS in the nucleus, but this does not alter stability of WUS. The authors do not explain why these two treatments, both increasing WUS concentration in nucleus, have different effects on WUS stability. Is it possible that DEX is less stable and after 24 h is not efficient in sending WUS to the nucleus? Testing control plants expressing eGFP-GR can help to answer this question.

Response: Thank you for this comment. As far as the effect of dex on 35S::eGFP-GR, the GFP translocated and accumulated stably in the nucleus even after 24h exposure showing that the effects are specific to WUS and dex was effective (see Rodriguez et al. 2016, PNAS 113 (41) pp. E6307-E6315; Fig. S1). Moreover, the degradation of 35S::eGFP-WUS-GR after 24 hrs dex treatment, was observed in the outer cell layers of the SAM while the rest of the tissue including the leaves stably accumulated the protein in the nucleus showing that Dex is active (Snipes et al. 2018; PLOS Genetics 14 (4) pp. e1007351).

The Dex treatment alone can translocate WUS into the nucleus but fails to build up in the nucleus due to the nuclear export and subsequent destabilization in the cytoplasm by using the EAR-like domain (see point#9 below). However, blocking nuclear export by using LEP-B keeps WUS in the nucleus leading to its stabilization. A similar stable nuclear accumulation was also observed by mutating the nuclear export signal (the EAR like domain)-35S::eGFP-WUS-EARLM-GR (Snipes et al. 2018; PLOS Genetics 14 (4) pp. e1007351) showing the role of EAR-like domain in promoting nuclear export and also drawing a correlation between nuclear accumulation and protein stability. Moreover, the cells in outer cell layers lack cytokinin signaling, which in the inner layers/RM, has been shown to act on the nuclear retention signal to enrich WUS in the nucleus and improve its stability (Snipes et al. 2018; PLOS Genetics 14 (4) pp. e1007351). Similar lack of nuclear enrichment of WUS was seen with misexpression of eGFP-WUS from the CLV3 promoter showing that observed effects are not specific to GR fusions (Perales et al. 2016, PNAS 113 (41) pp. E6298-E6306). Also, nuclear enrichment in outer cell layers can also be achieved by adding a potent NLS from another source (Perales et al. 2016, PNAS 113 (41) pp. E6298-E6306). It is also important to note that, an extensive structure-function analysis of WUS protein did not reveal a clear presence of the nuclear localization signal (Rodriguez et al. 2016, PNAS 113 (41) pp. E6307-E6315) suggesting that facilitated nuclear import may not play a significant role in shaping the WUS gradient. Taken together, these experiments show that just

providing more wild type WUS to outer cell layers is not sufficient to enrich WUS in the nucleus but the nuclear export must be inhibited.

5. Fig. 1 There are many additional explanations for the data besides CLV3 offsetting nuclear export of WUS at the posttranscriptional level. For example, the treatment with MCLV3 should increase expression of endogenous WUS. Formation of WUS - eGFP-WUS-GR dimers might stabilize eGFP-WUS-GR interaction with DNA which might lead to its retention in the nucleus. The authors should add cycloheximide to investigate whether observed changes are dependent on translation.

Response: We thank the reviewer for this comment. However, we wish to clarify that as opposed to the reviewers argument, MCLV3 treatment instantaneously decreases the expression of endogenous WUS expression (Figs. 1J-K, 2I-J; Pages 34-35) (Lee et al. 2019; *Plant Cell Rep.* (2019) Mar; 38 (3) pp. 311-319). Therefore, the endogenous WUS may not affect the behavior of the eGFP-WUS-GR protein. In this context, the cycloheximide (Cyc) experiment is not necessary and also exposure of plants to general inhibitors of translation such as Cyc for 24 hrs is not ideal and the data may not be interpretable. To specifically answer the question that CLV3 may not block nuclear export but promotes DNA binding. If such is the case, one would expect that WUS still should accumulate at higher levels in the nuclei of the L1 layer of *clv3-2* mutants, which is not the case. Yes, it is possible that besides offsetting nuclear export, CLV3 could affect upstream processes such as DNA-binding and dimerization which influences free WUS levels in the nucleoplasm. This requires understanding of the CLV3-mediated WUS protein modifications and their influence on DNA binding and dimerization. However, lack of this knowledge should not affect the conclusions drawn on the role of CLV3 in offsetting the nuclear export and in turn regulating diffusion into adjacent cells.

6. Page 6. “These observations show that the regulation of nuclear export is essential for maintaining the nuclear levels of WUS and suggest that CLV3-signaling may offset nuclear export.” The presented data show that WUS can move in and out of the nucleus. But the data does *not demonstrate [maybe suggest] that the nuclear export is regulated, much less demonstrating that this regulation is due to CLV3.*

Response: We thank the reviewer for this comment. There are two aspects here. First, our interpretations of nuclear levels of WUS is regulated by the nuclear export are based on several lines of solid experimental evidence. i) WUS binds to EXPORTIN protein which depends on the functional EAR-like domain. ii) The mutations in EAR-like domain lead to a higher nuclear accumulation in the case of WUS expressed from the native WUS promoter (pWUS::eGFP-WUS, see Rodriguez et al. 2016, PNAS 113 (41) pp. E6307-E6315) and also in the stable nuclear accumulation of 35S::eGFP-WUS-GR-EARLM upon dex treatment (see

Snipes et al. 2018; PLOS Genetics 14 (4) pp. e1007351). iii) The nuclear export inhibitor-LMB treatment also leads to increase in nuclear WUS levels.

Second, WUS protein fails to accumulate at proportionately higher levels in the nuclei of the L1 layer cells compared to the inner cell layers in *clv3-2* mutants. Several possible explanations are possible; i) a decrease in WUS transport from inner to the outer L1 cell layer, ii) increased nuclear export and increased degradation/diffusion. The LEP-B treatment can increase nuclear levels in the L1 layer of *clv3-2* mutants showing that protein can migrate to the L1 layer. Since LEP-B directly interferes with nuclear export, the increase in nuclear levels in *clv3-2* mutants shows that nuclear export regulation occurs downstream of the CLV3 signaling. This was tested further by providing CLV3 signaling (MCLV3) to *clv3-2* mutants which resulted in stable nuclear accumulation, taken together these observations suggest that CLV3 signaling offsets nuclear export to regulate nuclear levels. Now these aspects have been clearly explained in the revised section and use the word “suggests”. The earlier title “CLV3 signaling offsets nuclear export of WUS” changed to better reflect the observations presented in this section as “**WUS nuclear export regulation occurs downstream of CLV3 signaling**” (see Page 8, Line 151).

Since LEP-B treatments of *pWUS::eGFP-WUS* sequesters WUS in the nucleus, it indirectly influences cytoplasmic WUS and could inhibit its diffusion into overlying cells. That could lead to the variable accumulation in the L1 and L2 layers. As a consequence, though all 3 cell layers accumulated WUS at higher level, the statistical significance was observed for the L3 (Fig. 4; Page 37). Therefore, to overcome the experimental limitation imposed by the interlinked regulation, we used independent *35S::eGFP-WUS-GR* system that allows transient translocation of WUS into the nucleus of all cells in the SAM to measure the effects of LEP-B which revealed a stable nuclear accumulation of WUS.

However, LEP-B treatment of *clv3-2* expressing *pWUS::eGFP-WUS* revealed a statistically significant increase in the nuclear WUS in the L1 and the L2 layers which shows that high levels of the protein produced in deeper cell layers can migrate into outer layers, but needs inhibition of nuclear export to stably accumulate in the nucleus.

In addition to offsetting nuclear export CLV3 may also play a role in improving WUS protein stability in the nucleus. Since a positive correlation can be seen between nuclear enrichment and improvement in protein stability, uncoupling them experimentally is not possible. Therefore, the option of CLV3 signaling stabilizing the WUS in the nucleus has been kept open. Since *pWUS::eGFP-WUS* fails to accumulate at higher levels in the cytosol of the L1 layer cells in *clv3-2* mutants, it suggested either CLV3 could stabilize WUS in the cytoplasm or inhibit diffusion. Since *35S::eGFP-WUS-GR* system allows measurement of protein accumulation in the cytoplasm, we took advantage to quantify the protein levels and found that it remains unchanged in *clv3-2* mutants and upon MCLV3 treatment (see new quantification in Fig. S8; Page 53). Therefore, ruling out the possibility of CLV3

signaling playing a role in stabilizing the cytoplasmic WUS and favoring the hypothesis that CLV3 signaling may inhibit WUS protein diffusion into adjacent cells. CLV3-mediated inhibition of diffusion of WUS also provides an explanation for the expansion of the stem cell domain and meristem size in *clv3-2* mutants.

7. Page 7 “To distinguish between these possibilities, we challenged the cytoplasmically localized **WUS (p35S::eGFP-WUS-GR)** with exogenous application of MCLV3 (Fig. S5). Our results show that MCLV3 treatment did not change the cytoplasmic WUS protein levels in both wild type and *clv3-2*”. There is no **quantification of the data**. Moreover, based on the submitted images, in the basal L3 tissues eGFP-WUS-GR seems to be moving to the nucleus in response to MCLV3.

Response: We thank the reviewer for this comment. We have now provided the quantification (Fig. S8; Page 53) and (see point#6 in the previous section for discussion). With regard to the sporadic and weak accumulation of WUS in the nuclei of few RM cells upon MCLV3 (before dex treatment), we found that it is not a consistent behavior (Fig. S4; Page 47). Therefore, it could be due to the leakiness of 35S::eGFP-WUS-GR which can be expected especially in regions where 35S::eGFP-WUS-GR accumulates at higher levels in the RM. Note a similar sporadic nuclear accumulation in the RM of EAR-like mutants -35S::eGFP-WUS-[EARLM]-GR before Dex application which can be attributed to leakiness of the GR fusion (Fig. S6; Page 50). It is also important to note that, an extensive structure-function analysis of WUS protein did not reveal presence of the nuclear localization signal (Rodriguez et al. 2016, PNAS 113 (41) pp. E6307-E6315) suggesting that facilitated nuclear import may not play a significant role in shaping the WUS gradient. Therefore, a uniform nuclear import rate was used for all cell layers in the model. Rather we have found clear evidence for nuclear retention and nuclear export in regulating the WUS gradient. The following sentence has been added to the results section (Page 6, Line 108) “In rare cases, nuclear accumulation of eGFP-WUS-GR was observed upon MCLV3 treatment in DEX-untreated plants (Fig. S4A'-B') which could be due to the leakiness of the GR fusion.”

8. Fig S2. qRT-PCR should be performed instead of semi-quantitative PCR.

Response: We thank the reviewer for this comment. We have now provided RT-PCR data on WUS expression for all treatment conditions and at all time points (Figures 1,2 and Supplementary Figure 3; Pages 34,35, and 45, respectively). The RT-PCR data showing severe downregulation of *WUS* transcription at all time points which strengthens our conclusions on a post-transcriptional role of CLV3 in stabilizing the WUS protein.

9. Fig S3P. Can the authors comment on why mutation of nuclear export signal leads to WUS accumulation in the cytosol? Should it not be otherwise?

Response: We thank the reviewer for noticing this. This could be due to the second function of EAR-like domain in destabilizing the protein in the cytoplasm. This hypothesis prompted us to examine the cytoplasmic levels of EAR-like domain mutants of WUS. Since EAR-like domain also functions in nuclear export it is not possible to visualize and quantify this phenomenon in plants expressing EAR domain mutants (*pWUS::eGFP-WUS-EARLM*). Therefore, we used the *35S::eGFP-WUS-GR-EARLM* to measure cytoplasmic levels. Our quantification revealed that EAR-like domain mutants accumulate at a significantly higher level than the wild type *35S::eGFP-WUS-GR* suggesting that EAR-like domain (Fig. S6; Page 50), in addition to its role in nuclear export, is also required for WUS degradation in the cytoplasm. Such a coupled behavior where nuclear export signal also functions as a degron in the cytoplasm has been observed for Aryl hydrocarbon receptor, a ligand (2,3,7,8- tetrachlorodibenzo-*p*-dioxin (TCDD)) activated nuclear TF-AHR (Davarinis and Pollenz 1999, J Biol Chem. Oct 1; 274 (40) pp. 28708-15). New data has been presented in Fig. S6 (Page 50) and the results are documented in section (Page 7, Line 119) “WUSCHEL physically interacts with EXPORTIN proteins” and we have updated the Fig. 5A (Page 38) to include this aspect.

This is particularly a relevant piece in the puzzle for two reasons; 1) From system regulation point of view, it leads to a co-ordination between export and degradation which in turn is critical for regulating cytoplasmic levels and diffusion. A fraction of the *pWUS::eGFP-WUS-EARLM* expressing plants developed bigger SAMs as shown in our earlier study (See Fig. S2 F, G, H; Rodriguez et al. 2016, PNAS 113 (41) pp. E6307-E6315). A relatively stable cytoplasmic *pWUS::eGFP-WUS-EARLM* that diffuses directly into adjacent cells (the other fraction gets into the nucleus) could lead to such overproliferation of SAMs. If EAR-like domain acted only as NES, the mutations could have impeded WUS movement leading to smaller or terminated SAMs which was not the case. The importance of the dual role of the EAR-like domain has been discussed in the revised fourth paragraph in the discussion section. 2) The degradation of exported WUS in the cytoplasm will continually create space for utilization of “new WUS species” that arrive into the nuclei of the CZ cells from the RM to activate *CLV3* transcription, thus coupling *CLV3*-mediated transcriptional regulation with the post-translational regulation. This aspect has now been added to the third paragraph of “Discussion” section which strengthens the proposed model.

Specific comments related to the computational model:

Several parts of the model involve fit-for-purpose and nonstandard forms of equations that do not seem to derive from biochemical principles. I don't object to phenomenological models, but clear descriptions and sound justifications must be included.

Response: We thank the reviewer for this comment. Some mathematical terms in the model were nonstandard and we created that for fitting the experimental

observations and choosing the associated parameters more easily. We have provided more detailed descriptions and justifications in the following responses and in the revised manuscript as well.

1. It looks like the authors assumed that the lateral restriction of CLV3 expression is due to the inhibition of lateral WUS diffusion by CLV3. The model basically has a 'highway' in the center of the SAM with lateral barriers (soundwall?). How can CLV3 only inhibit the lateral diffusion? Does it specifically recognize the plasmodesmata in the lateral direction? Also, this crucial assumption about lateral inhibition is not included in the listed equations.

Response: We thank the reviewer for this comment. In our model the same CLV3 strength used for restricting lateral diffusion significantly blocked WUS movement into the outer cell layers suggesting that either CLV3 does not inhibit diffusion into the apical cell layers or require lower strength than what is needed for inhibiting lateral diffusion. It is conceivable that such observed differences in simulations may in part be required for restricting lateral diffusion of WUS based on the following observations. The transient depletion of CLV3 has been shown to result in a staggered centripetal expansion of CLV3 promoter activity leading to increase in stem cell numbers and meristem size (Reddy and Meyerowitz 2005, Science 28 Oct: 310 (5748) pp. 663-667) which suggests that WUS could enter into the centrally-located cells in the L1 and L2 layers and then diffuses laterally. Moreover, the cells in the L1 and the L2 layers that divide in anticlinal planes may contain higher numbers of primary plasmodesmata (PDs) than the non-dividing cell wall that separates L3 from L2, and L2 from L1 layers. Perhaps these anatomical differences contribute to higher rates of WUS diffusion laterally than into the apical cell layers. Therefore, a signal such as CLV3 may be required for inhibiting lateral diffusion of WUS. Future work on the analysis of receptor systems involved in perceiving CLV3 signal, PD distribution in SAMs and WUS protein modifications may provide insights into the regulation of WUS diffusion and stem cell maintenance. This aspect has been summarized in the new final paragraph in the discussion.

We would like to point out that this assumption about lateral diffusion was included in the original model by using the diffusion coefficient in the format of $\frac{D_W}{1+([CLV3]/k_{CLV}^2)^{n_6}}$ (Fig. S9 xii; Page 54). In the numerical simulations, this inhibition of diffusion was implemented between cells in the same layers, but not cells from different layers.

2. Fig S6. The equation for C_P has a nonstandard form of nonlinear function that lacks mechanistic basis. Why can't the biphasic regulation of CLV3 by WUS be modeled with multiplication of Hill functions like other gene regulations?

Response: We thank the reviewer for this comment. The way we model the hat-shape function for CLV3 production is by connecting two different Hill functions to make it continuous at the connecting point, such that we can easily determine the maximum, minimum and EC50 values for different branches. We created this

function to model the concentration-dependent manner of WUS regulation on the CLV3 expression: WUS inhibits CLV3 at high concentration and promotes CLV3 at low concentration, but fails to activate CLV3 again at too low concentration. The hat-shape function we used in the model can be replaced by a production of two Hill functions by choosing appropriate parameters, which is shown as below

In this figure, the function on the left is the one used in our model. The right function is a product of two Hill functions $\frac{2.5}{(1+(x/3)^{-6})(1+(x/6)^6)}$. They have very similar shapes. However, it is easier to choose parameters in the stepwise function to achieve a certain maximum and a flat region near the maximum. Both functions can be applied to model the transcription of CLV3 in the model by choosing parameters appropriately.

3. Most of the Hill coefficients are extremely high (10). What are the possible reasons for this high nonlinearity from a mechanistic point of view? For gene regulations this may be explained by high numbers of binding sites, but for processes like diffusion and nuclear transport, it's difficult to conceive the mechanisms.

Response: We thank the reviewer for this comment. The dual function of CLV3 in restricting WUS transcription to the L3 and in regulating the nuclear levels of WUS in the outer layers requires a switch like regulation, which is modeled by the high Hill coefficients in the model. It is possible that spatially restricted receptors or intracellular signaling networks that mediate the dual regulation may lead to highly nonlinear effects of CLV3 regulation in producing different outputs in such closely spaced domains. This explanation has been added to the new section 'Calibration of the Computational Model' in the revised manuscript (see Page 23, Line 490).

4. The parameter values for restricting expressing domains of WUS and CLV3 (e.g. radius) are not listed. The domain restriction of CK receptor is not clearly defined in the equations.

Response: We thank the reviewer for this comment. We have added the parameters about the expressing domains of WUS and CLV3, r_w and L_w , in Supplementary Table 1 (Page 65), as well as the formula of expression of CK receptor in the revised Supplementary Figure 9 (Page 54).

5. The threshold parameters in Hill functions are known as EC50, not EZ50.

Response: We thank the reviewer for detecting this typo. We have now corrected it in the revised Supplementary Table 1 (Page 65).

REVIEWER COMMENTS

Reviewer #1 (Remarks to the Author):

In my previous review, I focused on the technical approach of the authors regarding the computational model. Therefore, I will respond to the authors activities in this respect. As I wrote in my previous review I consider this paper as a very good example of a Systems Biology work. However, it had one large deficiency and in my opinion the authors only partially managed to remove it.

Using global sensitivity analysis methods is a very good idea to reduce the dimensionality of the uncertainty or the admissible parameter space. The authors improved their work considerably. Nevertheless, it remains unclear how EXACTLY the authors have chosen their parameters. It is clear from their description that certain experimental observations are matched. Was this ONLY possible by the parameter set they have chosen? Or are there other choices? I completely understand that exploring a 40 dimensional parameter space is challenging. But the way the authors present their approach is a bit unclear. The value of the local sensitivity analysis is obscured by the vagueness of their parameter choices. I do not believe that the parameter set the authors are operating with is the only possible choice. What does this mean to the (local) sensitivity analysis?

The authors make statements such as: "... a 25% lower WUS expression rate led to the internalized expression of CLV3 ...". 25 % not 24 % or 23 %?

What I mean is even after doing sensitivity analysis to reduce the dimensionality of the parameter space, taking observations to further constrain the parameter space, an uncertainty in the input-parameters of the computational model will remain. This will, inevitably lead to uncertainties in the model response. This does not render the model as useless or arbitrary. The model allows for conceptual statements and testing of perturbations. This is what the authors used the model for, but I suggest that the authors refrain from statements like "25 %..." which somewhat sound precise. The authors should rather use statements like or similar to: "for all parameter sets for which the model behaves according to the experimental observations the model perturbation of X leads to ...". This requires, though, an analysis of the admissible parameter space and not operating with only one parameter set.

One word about taking parameters from literature. Unless the parameters were estimated using the same model or the parameters describe precise bio-chemical or physical processes (like expression rates, protein stabilities, diffusion constants, protein-protein binding rates, etc.), using values from previous modelling approaches is very questionable.

In case the authors are convinced that the admissible parameter space is very small due to certain reasons such that a local analysis and local statements are justified, they need to justify it better in their text.

I would like to re-state my previous statement: I think this paper could be a very good and solid piece of Systems Biology and worthy to be published in NCOMM, but the standards need to be high not only on the experimental side of the work. The current version of the MS is improved over the first version but still does not meet the standard it should have regarding the modelling. In its current form I cannot recommend publication in NCOMM.

Reviewer #3 (Remarks to the Author):

In the revision, the authors addressed the majority of my concerns. They considerably expanded their analysis of pWUS::eGFP-WUS expression, added a control (p35S::eGFP-GR) and performed the requested statistical analysis. Their detailed rebuttal gives a satisfactory response to many of my questions. The new figures are clear and well-organized. Taken together, the data convincingly show an increased stability of nuclear WUS and the ability of CLV3 to stabilize WUS. I have only minor suggestions.

1. The focus on differences of WUS stabilization in different meristematic layers in the experimental section of the manuscript is distracting and sometimes misleading. E.g. the first result section is titled "Exogenous CLAVATA3 application increases WUSCHEL protein levels in the L1 layer". This is true for the *clv3* mutant, but not for the wt. And even in *clv3* the effect on the L1 layer is short-lived. But considering that in the presence of CLV3, WUS mRNA level goes down and WUS protein levels stay high, I agree with the main conclusion that CLV3 stabilizes WUS. I recommend looking over the manuscript and see where the focus on the layers is essential and where it is a distraction from the main message. I found it in most cases confusing.

2. I find the title of the paper to be vague and convoluted. I suggest changing it to a more straightforward and memorable title.

3. On graphs, asterisks representing statistical significance blend in with the points representing data. Consider changing the color or size of asterisks. Figure 2I: Please check the locations of asterisks. It seems that one asterisk for 72h is in the wrong position.

4. Fig.1 and 2 Please clarify in the legend whether RT-PCR is measuring the combined expression of the eGFP-WUS reporter and the endogenous WUS, or just of the reporter.

5. On the graphs it would be helpful to see y axes with tick marks.

REVIEWER COMMENTS

Reviewer #1 (Remarks to the Author):

In my previous review, I focused on the technical approach of the authors regarding the computational model. Therefore, I will respond to the authors activities in this respect. As I wrote in my previous review I consider this paper as a very good example of a Systems Biology work. However, it had one large deficiency and in my opinion the authors only partially managed to remove it.

Using global sensitivity analysis methods is a very good idea to reduce the dimensionality of the uncertainty or the admissible parameter space. The authors improved their work considerably. Nevertheless, it remains unclear how EXACTLY the authors have chosen their parameters. It is clear from their description that certain experimental observations are matched. Was this ONLY possible by the parameter set they have chosen? Or are there other choices? I completely understand that exploring a 40 dimensional parameter space is challenging. But the way the authors present their approach is a bit unclear. The value of the local sensitivity analysis is obscured by the vagueness of their parameter choices. I do not believe that the parameter set the authors are operating with is the only possible choice. What does this mean to the (local) sensitivity analysis?

Response: We thank the reviewer for this comment. We appreciate the reviewer's recognition of the challenges involved in exploring a high dimensional parameter space. We found four additional biologically relevant parameter sets from the global sensitivity analysis and ran perturbation studies for each of them. We now explain in more detail in the revised version of the paper how the parameter set used in the main text was chosen and also provided additional parameter sets to test the main mechanism. Below we provide explanation on the choice of the parameter sets, salient features of the selected parameter sets and the perturbation analysis of the additional parameter sets other than the one presented in earlier version of the manuscript. The new analysis and results are now described in sections on '**Calibration of the Computational Model**', '**Additional Parameter Sets to Verify Robustness Mechanism**' and '**Perturbation Studies using Additional Parameter Sets**' and the data has been provided in **Supplementary Figures 17-22**.

i) Basic PDE model with reduced complexity. In this work, we implemented the Quasi Monte Carlo simulations with Sobol sampling over a sufficiently large parameter space for a simpler model first. This simpler model did not have the extrinsic signals-CLV3 and CK, and only contained the framework describing the dynamics of WUS mRNA and protein in nucleus and cytoplasm at a cell-based structure. We reduced the number of parameters for the simpler model as below:

Considering the CK response and signaling are independent of the CLV3 and WUS levels (See reviewer#3; point #1 from previous rebuttal), the CK signaling was included only for benchmarking to choose appropriate parameter values that can produce the cytokinin mutant behavior and those values were fixed in this study. This reduced the parameter number to 25. Subsequently, the

global sensitivity analysis was applied to the *clv3* null mutant model by excluding both transcriptional (2 parameters) and post-translational regulation of CLV3 (4 parameters), as well as the CLV3 transcription (4 parameters) and diffusion (1 parameter) and degradation (1 parameter) of CLV3 peptides. The concentration dependent self-stabilization of nuclear WUS was excluded in the global sensitivity analysis (4 parameters) since WUS protein can be stabilized by improving the nuclear import which does not depend on CLV3 signaling (see Reviewer #3; comment #4 from previous rebuttal). The nuclear import rate of WUS protein was fixed because the biological experiments suggest nuclear import is not regulated by CLV3 and CK, and also no clear nuclear localization or import signal was detected in the WUS protein (See previous rebuttal reviewer #3; comments #4 and #7 from previous rebuttal). The parameters related to WUS synthesis domain were determined and fixed based on experimental images (2 parameters). Therefore, the remaining five critical parameters investigated in the global sensitivity analysis were the synthesis rate of WUS mRNA (A_1), diffusion of WUS protein (D_w), nuclear export rate of the WUS protein (r_{ex}), degradation rates of nuclear (d_{wn}) and cytoplasmic (r_{ex}) WUS.

This paragraph was added in the section on 'Calibration of the Computational Model'.

- ii) Selection of biologically relevant parameter sets:** As explained in the previous revision, we created N=220 samples following the Sobol Sampling method and performed a total of 2640 simulations to compute the sensitivity index of each parameter. The experimental data shows that
- The ratio of WUS nuclear protein between L3 and L1 was about 4 in *clv3* mutant and 3 in wild type (see Supplementary Figure 10h-i),
 - The NC ratio in each layer was about 1 in both wild type and *clv3* mutant (see Supplementary Figure 10j).
 - Nuclear WUS protein accumulates in all cell layers including the outer L1 and L2 layers.

We selected parameter sets satisfying the ratio of WUS nuclear protein between L3 and L1 to be between 3 and 6, and NC ratios in each layer to be between 0.5 and 2, as well as the nuclear WUS to be greater than some threshold level (chosen to be 1 in this study) in L1 and L2 layers. It was found that 4 out of 220 satisfied all these criteria. The other 4 satisfied the first two criteria about the ratios, however, they failed to accumulate WUS protein in the nuclei of the L1 and the L2 layers above the chosen threshold even without CLV3 mediated repression of *WUS* transcription. With the inclusion of CLV3-mediated repression of *WUS* transcription, the WUS protein levels fell further which made the system highly sensitive to obtain the balance between WUS and CLV3 in wild type. Therefore, we considered the four parameter sets that satisfied all three criteria for further calibration and perturbation studies.

This paragraph was added in the section on 'Additional Parameter Sets to Verify Robustness Mechanism'.

- iii) Selection of parameters associated with the WUS concentration dependent regulation of CLV3 expression and with the CLV3 signaling effects on the WUS gradient:**

- Parameters involved in wild type *CLV3* expression: $W_\theta^1, W_\theta^2, W_\theta^3$

The WUS nuclear levels in *clv3* mutants were used to determine the *CLV3* activation window as described below: The WUS nuclear levels in *clv3* mutants were used to estimate the WUS nuclear levels in wild type, which were then used to determine the *CLV3* activation peak (W_θ^2).

W_{θ}^1 determined the WUS concentration threshold below which *CLV3* expression reduces more than 50%. W_{θ}^3 determined the WUS concentration threshold above which *CLV3* expression reduces more than 50% (see Supplementary Figure 17c). The experimental evidence shows that the wild type L1 nuclear level of WUS is slightly repressive to *CLV3*, i.e., it falls at the right hand side of the *CLV3* activation peak (Supplementary Figure 17a-c) Perales et al., *PNAS* **113**,(2016).

2. Parameters involved in *CLV3* diffusion and degradation: D_c, d_c, A_2

These parameters determined the decay length of *CLV3* peptides, i.e., how deep *CLV3* peptides could diffuse (D_c) from the L1 and L2 layers. In particular, the decay length of a diffusive molecule was equal to $\sqrt{D_c/d_c}$. In the calibration, we chose them to satisfy that *CLV3* peptides reached at least L3 layers. In addition, degradation (d_c) together with the *CLV3* synthesis rate A_2 determined the maximal level that *CLV3* could reach, which is A_2/d_c . Therefore, we fixed D_c and A_2 , and perturb d_c only to study the effects of the *CLV3* decay length and the maximal level of *CLV3* on the WUS gradient.

3. Parameters involved in *CLV3* transcriptional and post-translational regulations: $k_{cw}^1, n_1, k_{cw}^2, n_6, k_{cw}^3, n_3$

The simulation results would be sensitive to k_{cw}^i for $i = 1,2,3$ and they were chosen appropriately such that *CLV3* regulations would take effect in the model. In particular, k_{cw}^i are EC50 values in the Hill functions modeling the *CLV3* regulations and they were chosen to be less than the maximal level of *CLV3* in order to generate an effective *CLV3* regulation. The Hill coefficients n_1, n_3, n_6 determined the nonlinearity of the regulations depending on the *CLV3* concentration, i.e., larger coefficients gave rise to more switch-like regulations. The dual function of *CLV3* in restricting *WUS* transcription to the L3 and in regulating the nuclear levels of WUS in the outer layers requires a switch-like regulation, therefore high Hill coefficients were chosen in the model as explained previously (see previous rebuttal reviewer #3; comment #3 related to the computational model in the previous rebuttal) .

By integrating the results of Quasi Monte Carlo sensitivity analysis and the steady state analysis of the corresponding ODE system along with the properties of *CLV3* transcription and regulations, we were able to calibrate the computational model and obtain an appropriate parameter set that could generate simulation results close to the experimental data based on the observations on the protein levels in both nucleus and cytoplasm at different layers, as shown in Figure 5.

This paragraph was added in the section on 'Calibration of the Computational Model'. A schematic diagram was added in Supplementary Figure 22.

iv) Perturbation studies on the additional parameter sets: Given that both experimental data and simulation results use arbitrary units, the chosen parameter sets may generate different absolute WUS protein levels while maintaining specified relative fold change between different cell layers and NC ratios measured in experiments. Therefore, as the reviewer suggested, in this revision, we performed perturbation studies on those four additional parameter sets to test the proposed mechanism.

The analysis revealed that when *CLV3* is transcribed in the biologically relevant repression window of WUS concentration (Supplementary Figure 17), the post-translational regulation along with the transcriptional regulation was able to maintain the normal *CLV3* expression domain in L1 and L2 layers over a large range of *WUS* mRNA production rates (Supplementary Figures 18-21). Our simulations also revealed that the *CLV3* expression domain could be maintained with

transcriptional regulation only when *CLV3* is transcribed within the activation window of *WUS* concentration (Supplementary Figures 18-21). Theoretically, within the activation window, higher *WUS* levels activate *CLV3* which in turn represses *WUS* transcription, whereas lower *WUS* levels lead to loss of *CLV3*-mediated repression which restores *WUS* expression. However, such a mechanism is not biologically relevant because the wild type *CLV3* expression is regulated within the repression window of *WUS* concentration.

This paragraph was added in the section on 'Perturbation Studies using Additional Parameter Sets'. The results were provided in Supplementary Figures 18-21.

The authors make statements such as: "... a 25% lower *WUS* expression rate led to the internalized expression of *CLV3* ...". 25 % not 24 % or 23 %?

What I mean is even after doing sensitivity analysis to reduce the dimensionality of the parameter space, taking observations to further constrain the parameter space, an uncertainty in the input-parameters of the computational model will remain. This will, inevitably lead to uncertainties in the model response. This does not render the model as useless or arbitrary. The model allows for conceptual statements and testing of perturbations. This is what the authors used the model for, but I suggest that the authors refrain from statements like "25 %..." which somewhat sound precise. The authors should rather use statements like or similar to: "for all parameter sets for which the models behaves according to the experimental observations the model perturbation of X leads to ...". This requires, though, an analysis of the admissible parameter space and not operating with only one parameter set.

Response: We thank the reviewer for this comment. We have rewritten the results of the perturbation study by replacing those specific values by more general context. We kept the "25% of decrease in the efficacy of *CLV3*" since it was an assumption and fixed in the model with spatial requirement on *CLV3* expression. We also added the following description of perturbation studies of additional appropriate parameters on Page 13:

"Perturbation studies performed on different biologically relevant parameter sets showed that at lower *WUS* expression levels, the dual regulation resists the internalization of *CLV3* better than the transcriptional regulation only thereby maintaining the *WUS* transcription and *WUS* gradient (Supplementary Figures 19, 21). At higher *WUS* expression levels, the dual model resists a severe repression of *CLV3* thereby preventing the overproduction of *WUS* (Supplementary Figures 18, 20, 21). Since the proposed mechanism improves robustness over different parameter sets highlights its importance in maintaining the *WUS* gradient and stem cell homeostasis (see Methods section for more details; Supplementary Figures 18-21)."

One word about taking parameters from literature. Unless the parameters were estimated using the same model or the parameters describe precise bio-chemical or physical processes (like expression rates, protein stabilities, diffusion constants, protein-protein binding rates, etc.), using values from previous modelling approaches is very questionable.

Response: We thank the reviewer for this comment. We only used two parameters described in our previous publication, which are the diffusion rate of *CLV3* and *WUS* mRNA degradation rate Yadav et al., *Mol. Syst. Biol.* **9**, 654 (2013). They were calibrated in the previous study and were

within biologically relevant range. In the current study, we used the similar equations for WUS mRNA and CLV3 signaling. The current model was improved by including the additional regulations of CLV3 on the post-translational processes and by using two different variables to represent WUS protein in different subcellular compartments. Since the diffusion rate of CLV3 and the degradation of *WUS* mRNA are independent of the regulations explored in the current model, their calibrated values are fixed. All other parameters were calibrated in the current work (see **Calibration of the Computational Model** for details).

In case the authors are convinced that the admissible parameter space is very small due to certain reasons such that a local analysis and local statements are justified, they need to justify it better in their text.

Response: We thank the reviewer for this comment. As suggested, we performed perturbation studies on the four additional parameter sets obtained in the global sensitivity analysis that could generate the WUS gradient consistent with the experimental quantification in *clv3* mutants. The new analysis shows that the proposed mechanism can provide robustness over a range of parameters, indicating that it is a general principle regulating the WUS gradient. The results and analysis using additional parameter sets are provided in the section on '**Perturbation Studies using Additional Parameter Sets**' in **Method** section.

Reviewer #3 (Remarks to the Author):

In the revision, the authors addressed the majority of my concerns. They considerably expanded their analysis of pWUS::eGFP-WUS expression, added a control (p35S::eGFP-GR) and performed the requested statistical analysis. Their detailed rebuttal gives a satisfactory response to many of my questions. The new figures are clear and well-organized. Taken together, the data convincingly show an increased stability of nuclear WUS and the ability of CLV3 to stabilize WUS. I have only minor suggestions.

1. The focus on differences of WUS stabilization in different meristematic layers in the experimental section of the manuscript is distracting and sometimes misleading. E.g. the first result section is titled "Exogenous CLAVATA3 application increases WUSCHEL protein levels in the L1 layer". This is true for the *clv3* mutant, but not for the wt. And even in *clv3* the effect on the L1 layer is short-lived. But considering that in the presence of CLV3, WUS mRNA level goes down and WUS protein levels stay high, I agree with the main conclusion that CLV3 stabilizes WUS. I recommend looking over the manuscript and see where the focus on the layers is essential and where it is a distraction from the main message. I found it in most cases confusing.

Response: We thank the reviewer for this comment. We have updated the title of this section as follows: "**Exogenous CLAVATA3 prolongs WUSCHEL protein accumulation despite rapid loss of expression**". We think that mentioning layers is important because *clv3-2* mutants still accumulate higher nuclear WUS in the L2 layers and below than the L1 layer. This could be due to a higher WUS synthesis in deeper cell layers of *clv3-2* mutants which suggests that CLV3 is not required for stabilizing nuclear WUS if it exceeds a concentration threshold required

for self-stabilization. The second and related point about CLV3-mediated nuclear stabilization of WUS being transient also suggests that CLV3 may only offset nuclear export slowing down degradation and may not inhibit nuclear degradation. The transient increase in nuclear WUS in wild type plants may be sufficient to fine tune *CLV3* transcription to regulate nuclear export rates. Therefore, a transient increase rather than an irreversible increase may provide a tunable mechanism.

2. I find the title of the paper to be vague and convoluted. I suggest changing it to a more straightforward and memorable title.

Response: We thank the reviewer for the suggestions. As suggested, we have improved the title. “**CLAVATA3 mediated layer specific simultaneous control of transcriptional and post-translational processes provide robustness to the WUSCHEL gradient.**”

3. On graphs, asterisks representing statistical significance blend in with the points representing data. Consider changing the color or size of asterisks. Figure 2I: Please check the locations of asterisks. It seems that one asterisk for 72h is in the wrong position.

Response: We thank the reviewer for suggestions on asterisks clarity and for detecting this typo. We have now corrected Figure 2I.

4. Fig.1 and 2 Please clarify in the legend whether RT-PCR is measuring the combined expression of the eGFP-WUS reporter and the endogenous WUS, or just of the reporter.

Response: We thank the reviewer for this comment. Our Rt-PCR analysis measures the combined expression of *eGFP-WUS* reporter and the endogenous *WUS*. As suggested, we have now updated the legends and the following sentence has been added to the methods section.

“Luna SYBR reagents (NEB) were used to perform quantitative RT-PCR of the combined expression of the endogenous *WUS* and *eGFP-WUS* reporter on the BioRad quantitative thermocycler.”

5. On the graphs it would be helpful to see y axes with tick marks.

Response: We thank the reviewer's suggestions. We have added ticks to the y axes.

REVIEWERS' COMMENTS

Reviewer #1 (Remarks to the Author):

In my previous reviews I insisted strongly that the authors improve and clarify the way the model is calibrated and are more transparent about the remaining uncertainties of the model predictions due to uncertainties in the model parameters.

I find now that the authors made a strong effort to remove these deficiencies of their work. I am happy with the manuscript, the way the model is presented, and how the subtleties of the model calibration is described.

I find the manuscript sufficiently improved and I recommend strongly its publication. I would like to thank the authors for their efforts!